# A chromatic feature detector in the retina signals visual context changes

Larissa Höfling[1,2], Klaudia P Szatko[1,2†], Christian Behrens[1], Yuyao Deng[1,2], Yongrong Qiu[1,2‡], David Alexander Klindt[3§], Zachary Jessen[4], Gregory W Schwartz[4], Matthias Bethge[2,5], Philipp Berens[1,2,5,6], Katrin Franke[1‡], Alexander S Ecker[7,8], Thomas Euler[1,2*]

[1]Institute for Ophthalmic Research, University of Tübingen, Tübingen, Germany; [2]Centre for Integrative Neuroscience, University of Tübingen, Tübingen, Germany; [3]SLAC National Accelerator Laboratory, Stanford University, Menlo Park, United States; [4]Feinberg School of Medicine, Department of Ophthalmology, Northwestern University, Chicago, United States; [5]Tübingen AI Center, University of Tübingen, Tübingen, Germany; [6]Hertie Institute for AI in Brain Health, Tübingen, Germany; [7]Institute of Computer Science and Campus Institute Data Science, University of Göttingen, Göttingen, Germany; [8]Max Planck Institute for Dynamics and Self-Organization, Göttingen, Germany

*For correspondence:
thomas.euler@cin.uni-tuebingen.de

Present address: [†]National Institute of Neurological Disorders and Stroke, National Institutes of Health, Bethesda, United States; [‡]Department of Ophthalmology, Byers Eye Institute, Stanford University School of Medicine, Stanford, United States; [§]Cold Spring Harbor Laboratory, Cold Spring Harbor, United States

Competing interest: The authors declare that no competing interests exist.

**Abstract** The retina transforms patterns of light into visual feature representations supporting behaviour. These representations are distributed across various types of retinal ganglion cells (RGCs), whose spatial and temporal tuning properties have been studied extensively in many model organisms, including the mouse. However, it has been difficult to link the potentially nonlinear retinal transformations of natural visual inputs to specific ethological purposes. Here, we discover a nonlinear selectivity to chromatic contrast in an RGC type that allows the detection of changes in visual context. We trained a convolutional neural network (CNN) model on large-scale functional recordings of RGC responses to natural mouse movies, and then used this model to search in silico for stimuli that maximally excite distinct types of RGCs. This procedure predicted centre colour opponency in transient suppressed-by-contrast (tSbC) RGCs, a cell type whose function is being debated. We confirmed experimentally that these cells indeed responded very selectively to Green-OFF, UV-ON contrasts. This type of chromatic contrast was characteristic of transitions from ground to sky in the visual scene, as might be elicited by head or eye movements across the horizon. Because tSbC cells performed best among all RGC types at reliably detecting these transitions, we suggest a role for this RGC type in providing contextual information (i.e. sky or ground) necessary for the selection of appropriate behavioural responses to other stimuli, such as looming objects. Our work showcases how a combination of experiments with natural stimuli and computational modelling allows discovering novel types of stimulus selectivity and identifying their potential ethological relevance.

## Editor's evaluation

This study presents a fundamental and very technically strong dataset of mouse ganglion cells responding to natural stimuli that include more natural chromatic properties. Fits of convolutional neural networks to experimental measurements highlighted a novel form of color opponency in suppressed-by-contrast ganglion cells. More generally, the work provides a compelling example of how modern experimental and computational tools can be used to generate and test hypotheses about sensory function under natural conditions.

## Introduction

Sensory systems evolved to generate representations of an animal's natural environment useful for survival and procreation (*Lettvin et al., 1959*). These environments are complex and high dimensional, and different features are relevant for different species (reviewed in *Baden et al., 2020*). As a consequence, the representations are adapted to an animal's needs: features of the world relevant for the animal are represented with enhanced precision, whereas less important features are discarded. Sensory processing is thus best understood within the context of the environment an animal evolved in and that it interacts with (reviewed in *Turner et al., 2019*; *Simoncelli and Olshausen, 2001*).

The visual system is well suited for studying sensory processing, as the first features are already extracted at its experimentally well-accessible front-end, the retina (reviewed in *Kerschensteiner, 2022*; *Baden et al., 2020*). In the mouse, this tissue gives rise to around 40 parallel channels that detect different features (*Goetz et al., 2022*; *Baden et al., 2016*; *Bae et al., 2018*; *Rheaume et al., 2018*), represented by different types of retinal ganglion cells (RGCs), whose axons send information to numerous visual centres in the brain (*Martersteck et al., 2017*). Some of these channels encode basic features, such as luminance changes and motion, that are only combined in downstream areas to support a range of behaviours such as cricket hunting in mice (*Johnson et al., 2021*). Other channels directly extract specific features from natural scenes necessary for specific behaviours. For instance, transient OFF α cells trigger freezing or escape behaviour in response to looming stimuli (*Münch et al., 2009*; *Yilmaz and Meister, 2013*; *Kim et al., 2020*; *Wang et al., 2021*).

For many RGC types, however, we lack an understanding of the features they encode and how these link to behaviour (*Schwartz and Swygart, 2020*). One reason for this is that the synthetic stimuli commonly used to study retinal processing fail to drive retinal circuits 'properly' and, hence, cannot uncover critical response properties triggered in natural environments. This was recently illustrated at the example of spatial nonlinear processing, which was found to be more complex for natural scenes than for simpler synthetic stimuli (*Karamanlis and Gollisch, 2021*). Such nonlinearities, which are crucial for the encoding of natural stimuli, cannot be captured by linear-nonlinear (LN) models of retinal processing, and several alternative methods have been proposed for the analysis of natural stimulus responses (reviewed in *Sharpee, 2013*).

One approach to modelling nonlinear visual processing is to train a convolutional neural network (CNN) to predict neuronal responses. This approach has gained popularity in recent years, both in the retina (*McIntosh et al., 2016*; *Maheswaranathan et al., 2023*; *Tanaka et al., 2019*; *Batty et al., 2017*) and in higher visual areas (*Yamins et al., 2014*; *Cadena et al., 2019*; *Ustyuzhaninov et al., 2024*). The resulting models, also referred to as 'digital twins', offer a number of analysis techniques that have been used to investigate, for example, the contributions of different interneurons to a cell's response (*Maheswaranathan et al., 2023*), or the effects of stimulus context (*Fu et al., 2024*; *Goldin et al., 2022*) and behavioural state (*Franke et al., 2022*) on neural coding. In particular, feature visualisations (*Olah et al., 2017*) can be used to quickly generate stimuli that would maximally excite the modelled neurons (*Walker et al., 2019*; *Bashivan et al., 2019*), which in turn can serve as interpretable short-hand descriptions of nonlinear neuronal selectivities. In visual cortex, the resulting *maximally exciting inputs* (MEIs) revealed more complex and diverse neuronal selectivities than were expected based on previous results obtained with synthetic stimuli and linear methods (*Walker et al., 2019*; *Bashivan et al., 2019*).

Here, we combined the power of CNN-based modelling with large-scale recordings from RGCs to investigate colour processing in the mouse retina under natural stimulus conditions. Colour is a salient feature in nature, and the mouse visual system dedicates intricate circuitry to the processing of chromatic information (*Szél et al., 1992*; *Joesch and Meister, 2016*; *Baden et al., 2013*; *Szatko et al., 2020*; *Khani and Gollisch, 2021*; *Mouland et al., 2021*). Studies using simple synthetic stimuli have revealed nonlinear and centre-surround (i.e. spatial) interactions between colour channels, but it is not clear how these are engaged in retinal processing of natural, temporally varying environments. Since mouse photoreceptors are sensitive to green and UV light (*Jacobs et al., 2004*), we recorded RGC responses to stimuli capturing the chromatic composition of natural mouse environments in these two chromatic channels. A model-guided search for MEIs in chromatic stimulus space predicted a novel type of chromatic tuning in tSbC RGCs, a type whose function is being debated (*Mani and Schwartz, 2017*; *Tien et al., 2015*; *Tien et al., 2022*).

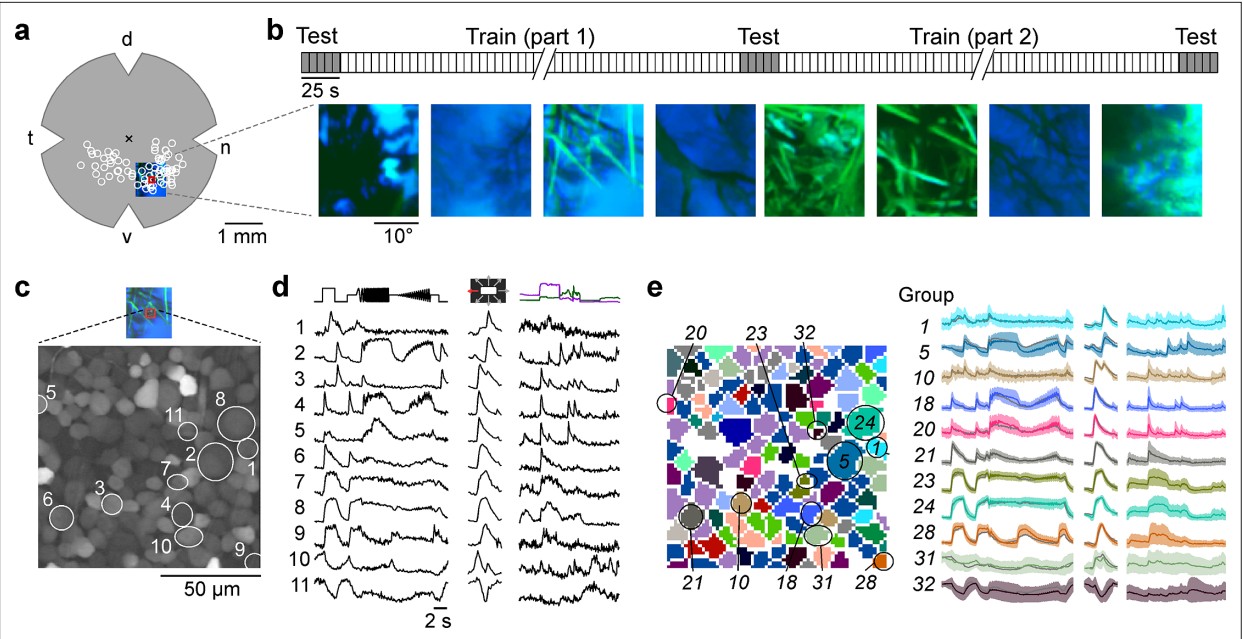

**Figure 1.** Mouse retinal ganglion cells (RGCs) display diverse responses to a natural movie stimulus. (**a**) Illustration of a flat-mounted retina, with recording fields (white circles) and stimulus area centred on the red recording field indicated (cross marks optic disc; d, dorsal; v, ventral; t, temporal; n, nasal). (**b**) Natural movie stimulus structure (top) and example frames (bottom). The stimulus consisted of 5 s clips taken from UV-green footage recorded outside (***Qiu et al., 2021***), with 3 repeats of a 5-clip test sequence (highlighted in grey) and a 108-clip training sequence (see Methods). (**c**) Representative recording field (bottom; marked by red square in (**a**)) showing somata of ganglion cell layer (GCL) cells loaded with $Ca^{2+}$ indicator OGB-1. (**d**) $Ca^{2+}$ responses of exemplary RGCs (indicated by circles in (**c**)) to chirp (left), moving bar (centre), and natural movie (right) stimulus. (**e**) Same recording field as in (**c**) but with cells colour-coded by functional RGC group (left; see Methods and ***Baden et al., 2016***) and group responses (coloured, mean ± SD across cells; trace of example cells in (**d**) overlaid in black).

The online version of this article includes the following figure supplement(s) for figure 1:

**Figure supplement 1.** Additional information about the dataset, model performance, and response quality filtering pipeline.

A detailed in silico characterisation followed up by experimental validation ex-vivo confirmed this cell type's pronounced and unique selectivity for dynamic full-field changes from green-dominated to UV-dominated scenes, a type of visual input that matches the scene statistics of transitions across the horizon (***Qiu et al., 2021***; ***Abballe and Asari, 2022***; ***Gupta et al., 2022***). We therefore suggest a role for tSbC RGCs in detecting behaviourally relevant changes in visual context, such as a transitions from ground (i.e. below the horizon) to sky (i.e. above the horizon).

## Results

Here, we investigated colour processing in the mouse retina under natural stimulus conditions. To this end, we trained a CNN model on RGC responses to a movie covering both achromatic and chromatic contrasts occurring naturally in the mouse environment, and then performed a model-guided search for stimuli that maximise the responses of RGCs.

### Mouse RGCs display diverse responses to a natural movie stimulus

Using two-photon population $Ca^{2+}$ imaging, we recorded responses from 8388 cells (in 72 recording fields across 32 retinae) in the ganglion cell layer (GCL) of the isolated mouse retina (***Figure 1a***) to a range of visual stimuli. Since complex interactions between colour channels have been mostly reported in the ventral retina and opsin-transitional zone, we focused our recordings on these regions (***Szatko et al., 2020***; ***Khani and Gollisch, 2021***).

The stimuli included two achromatic synthetic stimuli – a contrast and frequency modulation ('chirp' stimulus) and a bright-on-dark bar moving in eight directions ('moving bar', MB) – to identify the functional cell type (see below), as well as a dichromatic natural movie (***Figure 1b–d***). The latter was

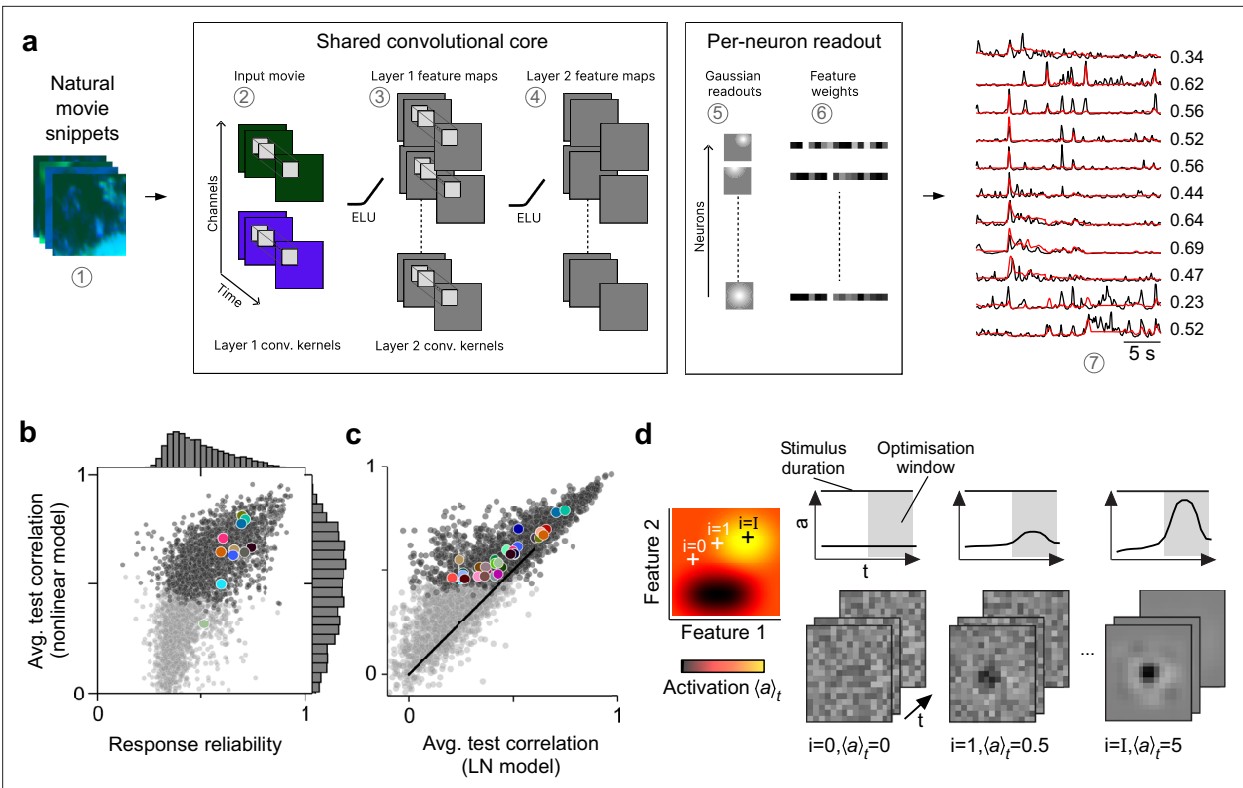

**Figure 2.** Convolutional neural network (CNN) model captures diverse tuning of retinal ganglion cell (RGC) groups and predicts maximally exciting inputs (MEIs). (**a**) Illustration of the CNN model and its output. The model takes natural movie clips as input (1), performs 3D convolutions with space-time separable filters (2) followed by a nonlinear activation function (ELU; 3) in two consecutive layers (2–4) within its core, and feeds the output of its core into a per-neuron readout. For each RGC, the readout convolves the feature maps with a learned RF modelled as a 2D Gaussian (5), and finally feeds a weighted sum of the resulting vector through a softplus nonlinearity (6) to yield the firing rate prediction for that RGC (7). Numbers indicate averaged single-trial test set correlation between predicted (red) and recorded (black) responses. (**b**) Test set correlation between model prediction and neural response (averaged across three repetitions) as a function of response reliability (see Methods) for N=3527 RGCs. Coloured dots correspond to example cells shown in *Figure 1c–e*. Dots in darker grey correspond to the N=1947 RGCs that passed the model test correlation and movie response quality criterion (see Methods and *Figure 1—figure supplement 1*). (**c**) Test set correlation (as in (**b**)) of CNN model vs. test set correlation of an LN model (for details, see Methods). Coloured dots correspond to means of RGC groups 1–32 (*Baden et al., 2016*). Dark and light grey dots as in (**b**). (**d**) Illustration of model-guided search for MEIs. The trained model captures neural tuning to stimulus features (far left; heat map illustrates landscape of neural tuning to stimulus features). Starting from a randomly initialised input (second from left; a 3D tensor in space and time; only one colour channel illustrated here), the model follows the gradient along the tuning surface (far left) to iteratively update the input until it arrives at the stimulus (bottom right) that maximises the model neuron's activation within an optimisation time window (0.66 s, grey box, top right).

composed of footage recorded outside in the field using a camera that captured the spectral bands (UV and green; *Qiu et al., 2021*) to which mouse photoreceptors are sensitive ($\lambda_{peak}^{S} = 360$, $\lambda_{peak}^{M} = 510$ nm for S- and M-cones, respectively; *Jacobs et al., 2004*). We used 113 different movie clips, each lasting 5 s, that were displayed in pseudo-random order. Five of these constituted the test set and were repeated three times: at the beginning, in the middle, and at the end of the movie presentation, thereby allowing to assess the reliability of neuronal responses across the recording (*Figure 1b*, top).

The responses elicited by the synthetic stimuli and the natural movie were diverse, displaying ON (*Figure 1d*, rows 4–9), ON-OFF (row 3), and OFF (rows 1 and 2), as well as sustained and transient characteristics (e.g. rows 8 and 4, respectively). Some responses were suppressed by temporal contrast (generally, rows 10, 11; at high contrast and frequency, row 9). A total of 6984 GCL cells passed our response quality criteria (see Methods); 3527 cells could be assigned to 1 of 32 previously characterised functional RGC groups (*Baden et al., 2016*) based on their responses to the chirp and MB stimuli using our recently developed classifier (*Figure 1e*; *Figure 1—figure supplement 1*; *Qiu et al., 2021*). Cells assigned to any of groups 33–46 were considered displaced amacrine cells and were not analysed in this study (for detailed filtering pipeline, see *Figure 1—figure supplement 1c*).

## CNN model captures diverse tuning of RGC groups and predicts MEIs

We trained a CNN model on the RGCs' movie responses (*Figure 2a*) and evaluated model performance as the correlation between predicted and trial-averaged measured test responses, $C(\hat{r}^{(n)}, \langle r^{(n)}\rangle_i)$ (*Figure 2b*). This metric can be interpreted as an estimate of the achieved fraction of the maximally achievable correlation (see Methods). The mean correlation per RGC group ranged from 0.32 ($G_{14}$) to 0.79 ($G_{24}$) (*Figure 1—figure supplement 1*) and reached an average of 0.48 (for all *N*=3527 cells passing filtering steps 1–3, *Figure 1—figure supplement 1*). We also tested the performance of our nonlinear model against a linearised version (see Methods; equivalent to an LN model, and from here on 'LN model') and found that the nonlinear CNN model achieved a higher test set correlation for all RGC groups (average correlation LN model: 0.38; $G_{14}$: 0.2, $G_{24}$: 0.65, *Figure 2c*).

Next, we wanted to leverage our nonlinear CNN model to search for potentially nonlinear stimulus selectivities of mouse RGC groups. Towards this goal, we aimed to identify stimuli that optimally drive RGCs of different groups. For linear systems, the optimal stimulus is equivalent to the linear filter and can be identified with classical approaches such as reverse correlation (*Chichilnisky, 2001*). However, since both the RGCs and the CNN model were nonlinear, a different approach was necessary. Other recent modelling studies in the visual system have leveraged CNN models to predict static MEIs for neurons in monkey visual area V4 (*Bashivan et al., 2019*; *Willeke et al., 2023*) and mouse visual area V1 (*Walker et al., 2019*; *Franke et al., 2022*). We adopted this approach to predict dynamic (i.e. time-varying) MEIs for mouse RGCs. We used gradient ascent on a randomly initialised, contrast- and range-constrained input to find the stimulus that maximised the mean activation of a given model neuron within a short time window (0.66 s; see Methods; *Figure 2d*).

It is important to note that MEIs should not be confused with, or interpreted as, the linear filters derived from classical approaches such as reverse correlation (*Chichilnisky, 2001*; *Schwartz et al., 2006*). While both MEIs and linear filters offer simplified views of a neuron's nonlinear response properties, they emphasise different aspects. The linear filter is optimised to provide the best possible linear approximation of the response function, identifying the stimulus direction to which the cell is most sensitive on average across the stimulus ensemble. In contrast, the MEI maximises the neuron's response by finding the single stimulus that activates the cell most strongly. Consequently, MEIs can differ significantly from linear filters, often exhibiting greater complexity and higher frequency components (*Walker et al., 2019*).

## MEIs reflect known functional RGC group properties

The resulting MEIs were short, dichromatic movie clips; their spatial, temporal, and chromatic properties and interactions thereof are best appreciated in lower-dimensional visualisations (*Figure 3a–c*; more example MEIs in *Figure 3—figure supplement 1*).

To analyse the MEIs in terms of these properties, we decomposed them into their spatial and temporal components, separately for green and UV, and parameterised the spatial component as a difference-of-Gaussians (DoG) (*Gupta et al., 2022*) (*N*=1613 out of 1947, see Methods). We then located MEIs along the axes in stimulus space corresponding to three properties: centre size, mean temporal frequency, and centre contrast, separately for green and UV (*Figure 3d–f*). These MEI properties reflect RGC response properties classically probed with synthetic stimuli, such as spots of different sizes (*Goetz et al., 2022*), temporal frequency modulations (*Baden et al., 2016*), and stimuli of varying chromatic contrast (*Szatko et al., 2020*; *Khani and Gollisch, 2021*). Using the MEI approach, we were able to reproduce known properties of RGC groups (*Figure 3g–i*). For example, sustained ON $\alpha$ RGCs ($G_{24}$), which are known to prefer large stimuli (*Baden et al., 2016*; *Mani and Schwartz, 2017*), had MEIs with large centres ($G_{24}$, *N*=20 cells: green centre size, mean ± SD: 195 ±82 μm; UV centre size 178 ±45 μm; average across all RGC groups: green 148 ±42 μm, UV 141 ±42 μm; see *Figure 3g*).

The MEI's temporal frequency relates to the temporal frequency preference of an RGC: MEIs of $G_{20}$ and $G_{21}$, termed ON high frequency and ON low frequency (*Baden et al., 2016*), had high and low average temporal frequency, respectively ($G_{20}$, *N*=40 cells, green, mean ± SD: 2.71 ±0.16 Hz, UV 2.86 ±0.22 Hz; $G_{21}$, *N*=50 cells, green, mean ± SD: 2.32 ±0.63 Hz, UV 1.98 ± 0.5 Hz; see *Figure 3h*). Some MEIs exhibit fast oscillations (*Figure 3e* and *Figure 3—figure supplement 1*). This is not an artefact but rather a consequence of optimising a stimulus to maximise activity over a 0.66 s time window (*Figure 2d*). To maximise the response of a transient RGC over several hundred milliseconds, it has to

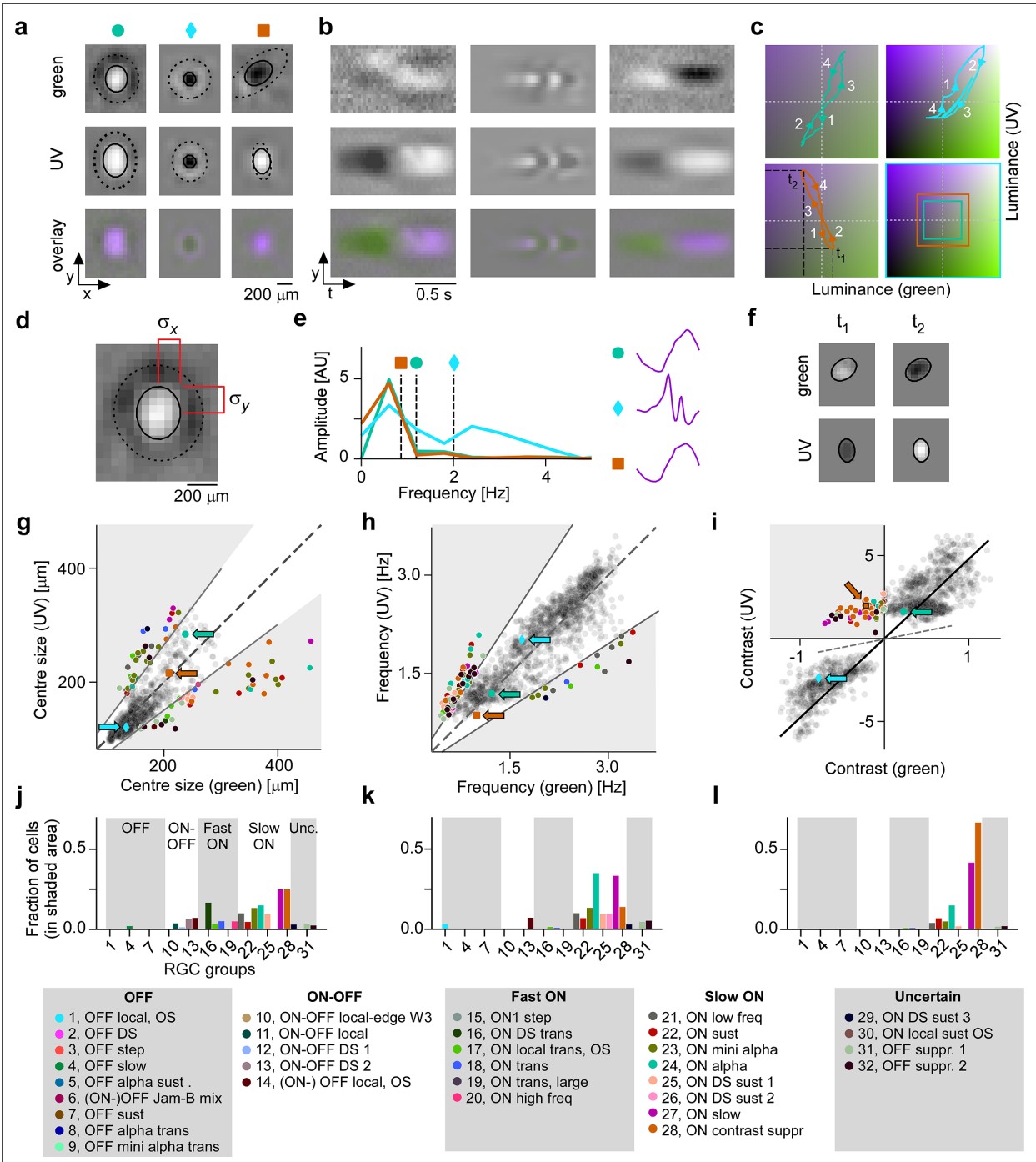

**Figure 3.** Spatial, temporal, and chromatic properties of maximally exciting inputs (MEIs) differ between retinal ganglion cell (RGC) groups. (**a**) Spatial component of three example MEIs for green (top), UV (middle), and overlay (bottom). Solid and dashed circles indicate MEI centre and surround fit, respectively. For display, spatial components $s$ in the two channels were re-scaled to a similar range and displayed on a common grey-scale map ranging from black for $-max(|s|)$ to white for $max(|s|)$, i.e., symmetric about 0 (grey). (**b**) Spatiotemporal ($y$–$t$) plot for the three example MEIs (from (**a**)) at a central vertical slice for green (top), UV (middle), and overlay (bottom). Grey-scale map analogous to (**a**). (**c**) Trajectories through colour space over time for the centre of the three MEIs. Trajectories start at the origin (grey level); direction of progress indicated by arrow heads. Bottom right: Bounding boxes of the respective trajectory plots. (**d**) Calculation of MEI centre size, defined as $\sigma_x + \sigma_y$, with $\sigma_x$ and $\sigma_y$ the s.d. in horizontal and vertical direction, respectively, of the difference-of-Gaussians (DoG) fit to the MEI. (**e**) Calculation of MEI temporal frequency: Temporal components are transformed using fast Fourier transform, and MEI frequency is defined as the amplitude-weighted average frequency of the Fourier-transformed temporal component. (**f**) Calculation of centre contrast, which is defined as the difference in intensity at the last two peaks (indicated by $t_1$ and $t_2$, respectively, in (**c**)). For the example cell (orange markers and lines), green intensity decreases, resulting in OFF contrast, and UV intensity increases, resulting in ON contrast. (**g**)

*Figure 3 continued on next page*

*Figure 3 continued*

Distribution of green and UV MEI centre sizes across *N*=1613 cells (example MEIs from (**a–c**) indicated by arrows; symbols as shown on top of (**a**)). 95% of MEIs were within an angle of ±8° of the diagonal (solid and dashed lines); MEIs outside of this range are coloured by cell type. (**h**) As (**g**) but for distribution of green and UV MEI temporal frequency. 95% of MEIs were within an angle of ±11.4° of the diagonal (solid and dashed lines). (**i**) As (**g**) but for distribution of green and UV MEI centre contrast. MEI contrast is shifted away from the diagonal (dashed line) towards UV by an angle of 33.2° due to the dominance of UV-sensitive S-opsin in the ventral retina. MEIs at an angle >45° occupy the upper left, colour-opponent (UV$^{ON}$-green$^{OFF}$) quadrant. (**j, k**) Fraction of MEIs per cell type that lie outside the angle about the diagonal containing 95% of MEIs for centre size and temporal frequency. Broad RGC response types indicated as in *Baden et al., 2016*. (**l**) Fraction of MEIs per cell type in the upper-left, colour-opponent quadrant for contrast.

The online version of this article includes the following figure supplement(s) for figure 3:

**Figure supplement 1.** Example maximally exciting inputs (MEIs) for example cell types.

**Figure supplement 2.** Illustration of how different time windows for optimisation affect maximally exciting input (MEI) temporal properties.

be stimulated repetitively, hence the oscillations in the MEI. Maximising the response over a shorter time period results in MEIs without oscillations (*Figure 3—figure supplement 2*).

Finally, the contrast of an MEI reflects what is traditionally called a cell's ON vs. OFF preference: MEIs of ON and OFF RGCs had positive and negative contrasts, respectively (*Figure 3i*). An ON-OFF preference can be interpreted as a tuning map with two optima – one in the OFF- and one in the ON-contrast regime. For an ON-OFF cell, there are hence two stimuli that are approximately equally effective at eliciting responses from that cell. Consequently, for the ON-OFF RGC groups, optimisation resulted in MEIs with ON or OFF contrast, depending on the relative strengths of the two optima and on the initial conditions (*Figure 3—figure supplement 1*, G$_{10}$, and see Discussion).

MEIs were also largely consistent within functional RGC groups (*Figure 3—figure supplement 1*). Where this was not the case, the heterogeneity of MEIs could be attributed to a known heterogeneity of cells within that group. For example, MEIs of G$_{31}$ RGCs were diverse (*Figure 3—figure supplement 1*), and the cells that were originally grouped to form G$_{31}$ probably spanned several distinct types, as suggested by the group's unusually high coverage factor (*Baden et al., 2016*). Together, these results provided strong evidence that RGCs grouped based on responses to synthetic stimuli (chirp and MB) also form functional groups in natural movie response space.

## CNN model predicts centre colour opponency in RGC group G$_{28}$

Our goal was to explore chromatic tuning of RGCs and to identify novel stimulus selectivities related to chromatic contrast. Therefore, we specifically focused on regions in stimulus space where a given stimulus property differs for green and UV. For centre size and temporal frequency, we asked which RGC groups contributed to the MEIs outside of the 95th percentile around the diagonal (*Figure 3g, h, j, and k*). These 5% MEIs furthest away from the diagonal were almost exclusively contributed by ON cells; and among these, more so by slow than by fast ON cells.

MEI contrast needed to be analysed differently than size and temporal frequency for two reasons. First, due to the dominance of UV-sensitive S-opsin in the ventral retina (*Szél et al., 1992*), stimuli in the UV channel were much more effective at eliciting RGC responses. As a result, the contrast of most MEIs is strongly shifted towards UV (*Figure 3i*). Second, contrast in green and UV can not only vary along positive valued axes (as is the case for size and temporal frequency), but can also take on opposite signs, resulting in colour-opponent stimuli. Whereas most MEIs had the same contrast polarity in both colour channels (i.e. both ON or OFF, *Figure 3c*, blue and turquoise trajectories), some MEIs had opposing contrast polarities in UV and green (*Figure 3c*, orange trajectory, and *Figure 3i*, upper left quadrant). Thus, for contrast, we asked which RGC groups contributed to colour-opponent MEIs (i.e. MEIs in the colour-opponent, upper left or lower right quadrant in *Figure 3i*). Again, slow ON RGCs made up most of the cells with colour-opponent MEIs. Here, G$_{28}$ stood out: 66% (24/36) of all cells of this group had colour-opponent MEIs (UV$^{ON}$-green$^{OFF}$), followed by G$_{27}$ with 42% colour-opponent MEIs.

The colour opponency we found in G$_{28}$ was not centre-surround, as described before in mice (*Szatko et al., 2020*), but rather a centre opponency ('co-extensive' colour-opponent RF; reviewed in *Schwartz, 2021*), as can be seen in the lower-dimensional visualisations (*Figure 3a and b*, right column; *Figure 3c*, orange trajectory).

In conclusion, our model-guided in silico exploration of chromatic stimulus space revealed a variety of preferred stimuli that captured known properties of RGC groups, and revealed a preference of G$_{28}$

RGCs for centre colour-opponent, UV$^{ON}$-green$^{OFF}$ stimuli, a feature previously unknown for this RGC group.

## Experiments confirm selectivity for chromatic contrast

Next, we verified experimentally that the MEIs predicted for a given RGC group actually drive cells of that group optimally. To this end, we performed new experiments in which we added to our battery of stimuli a number of MEIs chosen according to the following criteria: We wanted the MEIs to (i) span the response space (ON, ON-OFF, OFF, transient, sustained, and contrast-suppressed) and (ii) to represent both well-described RGC types, such as $\alpha$ cells (i.e. G$_{5,24}$), as well as poorly understood RGC types, such as suppressed-by-contrast cells (G$_{28,31,32}$) (*Figure 4a*). We therefore chose MEIs of RGCs from groups G$_1$ (OFF local), G$_5$ (OFF $\alpha$ sustained), G$_{10}$ (ON-OFF local-edge), G$_{18}$ (ON transient), G$_{20}$ (ON high frequency), G$_{21}$ (ON low frequency), G$_{23}$ (ON mini $\alpha$), G$_{24}$ (sustained ON $\alpha$), G$_{28}$ (ON contrast suppressed), G$_{31}$ (OFF suppressed 1), and G$_{32}$ (OFF suppressed 2). For simplicity, in the following we refer to the MEI of an RGC belonging to group $g$ as group $g$'s MEI, or MEI $g$.

We presented these MEIs on a regularly spaced 5 × 5 grid to achieve approximate centring of stimuli on RGC RFs in the recording field (*Figure 4b and c*). For these recordings, we fit models whose readout parameters allowed us to estimate the RGCs' RF locations. We used these RF location estimates to calculate a spatially weighted average of the responses to the MEIs displayed at different locations, weighting the response at each location proportional to the RF strengths at those locations (*Figure 4b*, red highlight, and *Figure 4d*, top). We then performed the same experiment in silico, confirming that the model accurately predicts responses to the MEIs (*Figure 4d*, bottom; *Figure 4—figure supplement 1*). These experiments allowed us to evaluate MEI responses at the RGC group level (*Figure 4e and f*; *Figure 3—figure supplement 1*).

We expected RGCs to show a strong response to their own group's MEI, a weaker response to the MEIs of functionally related groups, and no response to MEIs of groups with different response profiles. Indeed, most RGC groups exhibited their strongest (G$_{5,20,21,28,32}$) or second-strongest (G$_{1,10,23}$) response to their own group's MEI (*Figure 4g*, top). Conversely, RGC groups from opposing regions in response space showed no response to each others' MEIs (e.g. G$_{1,5}$ [OFF cells] vs. G$_{21-28}$ [slow ON cells]). The model's predictions showed a similar pattern (*Figure 4g*, bottom), thereby validating the model's ability to generalise to the MEI stimulus regime.

Notably, G$_{28}$ RGCs responded very selectively to their own MEI 28, displaying only weak responses to most other MEIs (*Figure 4f and g*, selectivity index G$_{28}$ to MEI 28, SI$_{G_{28}}$(28), defined as the average difference in response between MEI 28 and all other MEIs in units of standard deviation of the response, mean $\pm$ SD: $2.58 \pm 0.76$; see Methods). This was in contrast to other RGC groups, such as G$_{23}$ and G$_{24}$, that responded strongly to MEI 28, but also to other MEIs from the slow ON response regime (*Figure 4g*, top; *Figure 4—figure supplement 1*, SI$_{G_{23}}$(28), mean $\pm$ SD: $1.04 \pm 0.69$, SI$_{G,24}$(28), mean $\pm$ SD: $1.01 \pm 0.46$). Hence, our validation experiments confirm the model's prediction that RGC group G$_{28}$ is selective for centre colour-opponent, UV$^{ON}$-green$^{OFF}$ stimuli.

## G$_{28}$ corresponds to the tSbC RGC type

Next, we sought to identify which RGC type G$_{28}$ corresponds to. In addition to its unique centre colour opponency, the responses of G$_{28}$ displayed a pronounced transient suppression to temporal contrast modulations (chirp response in *Figure 1e*). Therefore, we hypothesised that G$_{28}$ corresponds to the tSbC RGC type (*Tien et al., 2015*; *Tien et al., 2016*; *Tien et al., 2022*), which is one of the three SbC RGC types identified so far in the mouse and is also referred to as ON delayed (OND) because of its delayed response onset (*Jacoby and Schwartz, 2018*).

To test this hypothesis, we performed cell-attached electrophysiology recordings (*Figure 5*) targeting tSbC/OND cells (*N*=4), identified by their responses to spots of multiple sizes (*Goetz et al., 2022*), and later confirmed by their distinctive morphology (*Jacoby and Schwartz, 2018*; type 73 in *Bae et al., 2018*; *Figure 5c and d*). We recorded spikes while presenting the MEI stimuli (*Figure 5a*, top). Just like G$_{28}$ RGCs in the Ca$^{2+}$ imaging, tSbC/OND cells exhibited a pronounced selectivity for MEI 28, and were suppressed by most other MEIs (*Figure 5a*, middle and bottom). Notably, the characteristic delayed response onset was visible in both the Ca$^{2+}$ (*Figure 4f*, top) and electrical (*Figure 5a*) responses but was not predicted by the model (*Figure 4f*, bottom).

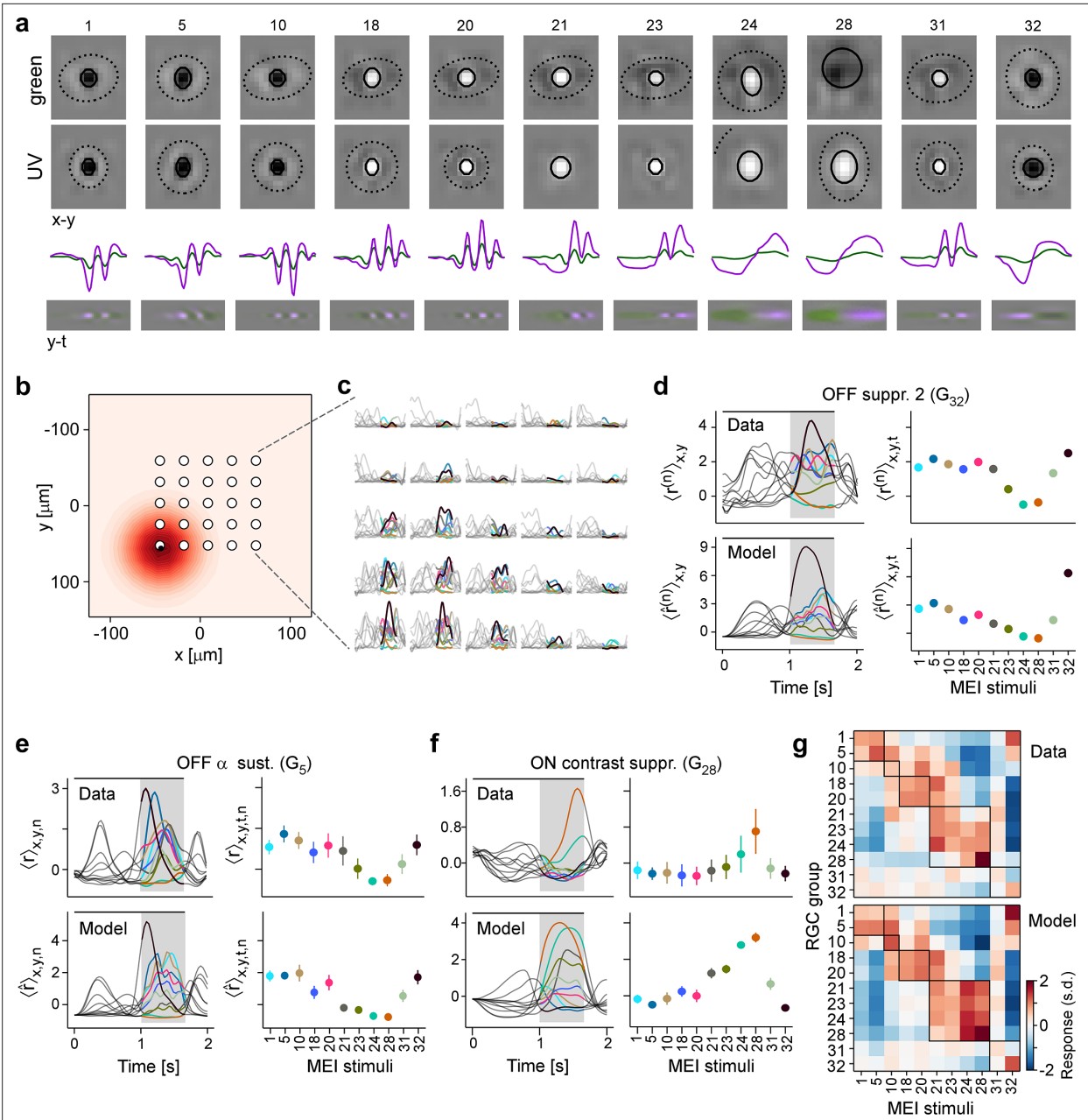

**Figure 4.** Experiments confirm maximally exciting inputs (MEIs) predicted by model. (**a**) MEIs shown during the experiment, with green and UV spatial components (top two rows), as well as green and UV temporal components (third row) and a spatiotemporal visualisation (fourth row). For display, spatial components $s$ in the two channels were re-scaled to a similar range and displayed on a common grey-scale map ranging from black for $-max(|s|)$ to white for $max(|s|)$, i.e., symmetric about 0 (grey). Relative amplitudes of UV and green are shown in the temporal components. (**b**) Illustration of spatial layout of MEI experiment. White circles represent $5 \times 5$ grid of positions where MEIs were shown; red shading shows an example RF estimate of a recorded $G_{32}$ retinal ganglion cell (RGC), with black dot indicating the RF centre position (Methods). (**c**) Responses of example RGC from (**b**) to the 11 different MEI stimuli at 25 different positions. (**d**) Recorded [top, $r^{(n)}$] and predicted [bottom, $\hat{r}^{(n)}$] responses to the 11 different MEIs for example RGC $n$ from (**b, c**). Left: Responses are averaged across the indicated dimensions $x$, $y$ (different MEI locations); black bar indicates MEI stimulus duration (from 0 to 1.66 s), grey rectangle marks optimisation time window (from 1 to 1.66 s). Right: Response to different MEIs, additionally averaged across time ($t$; within optimisation time window). (**e, f**) Same as in (**d**), but additionally averaged across all RGCs ($n$) of $G_5$ ($N$=6) (**e**) and of $G_{28}$ ($N$=12) (**f**). Error bars show SD across cells. (**g**) Confusion matrix, each row showing the z-scored response magnitude of one RGC group (averaged across all RGCs of that group) to the MEIs in (**a**). Confusion matrix for recorded cells (top; 'Data') and for model neurons (bottom; 'Model'). Black squares highlight broad RGC response types according to **Baden et al., 2016**: OFF cells, ($G_{1,5}$) ON-OFF cells ($G_{10}$), fast ON cells ($G_{18,20}$), slow ON ($G_{21,23,24}$) and ON contrast suppressed ($G_{28}$) cells, and OFF suppressed cells ($G_{31,32}$).

The online version of this article includes the following figure supplement(s) for figure 4:

*Figure 4 continued on next page*

*Figure 4 continued*

**Figure supplement 1.** Recorded and predicted responses of example RGC groups to the MEI stimuli.

As a control, we also recorded MEI responses of a different, well-characterised RGC type, sustained (s) ON $\alpha$ (G$_{24}$; *Krieger et al., 2017*; *Figure 5b*, top; *N*=4). Again, the electrical recordings of the cells' MEI responses yielded virtually the same results as the Ca$^{2+}$ imaging (*Figure 5b*, middle and bottom; *Figure 4—figure supplement 1*). Crucially, sON $\alpha$ cells were not selective for MEI 28. The fact that these experiments with precise positioning of stimuli on the cells' RFs yielded the same results as the 2P imaging experiments confirms the validity of the grid approach for stimulus presentation used in the latter.

## Chromatic contrast selectivity derives from a nonlinear transformation of stimulus space

Next, we asked whether G$_{28}$ (tSbC) RGC's selectivity is a linear feature, as could be achieved by two linear filters with opposite signs for the two colour channels, or whether it is a nonlinear feature. To address this question, we tested whether an LN model (implemented using convolutions; see Methods) could recover the chromatic selectivity of G$_{28}$ by predicting MEIs using the LN model. We found that the LN model predicted colour-opponent MEIs for only 9 out of 36 (25%) G$_{28}$ RGCs (nonlinear CNN: 24 out of 36 [66%] colour-opponent MEIs; *Figure 6a–c*). This finding argues against the possibility that G$_{28}$'s colour opponency can be explained on the computational level by two opposite-sign linear filters operating on the two colour channels, which could be recovered by an LN model. Instead, it suggests the presence of a nonlinear dependency between chromatic contrast (of the stimulus) and chromatic selectivity (of the cell). In other words, G$_{28}$ RGCs process stimuli differently depending on their chromatic contrast. This is a nonlinear property that cannot be accurately captured by an LN model that makes a single estimate of the linear filter for the whole stimulus space.

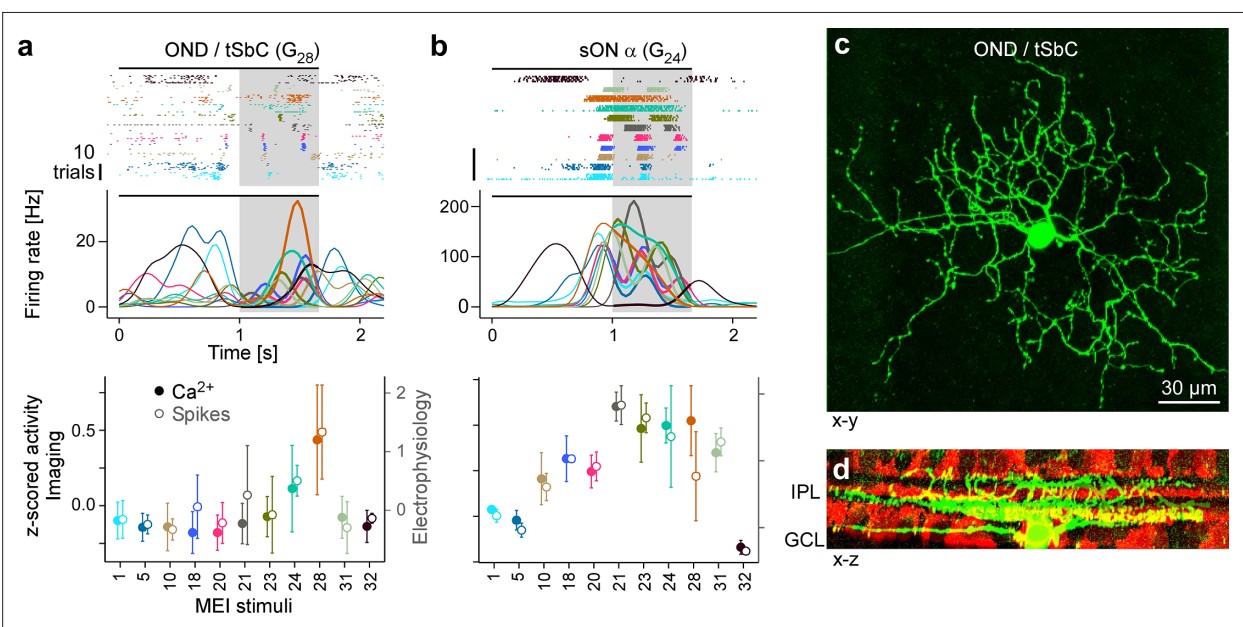

**Figure 5.** Electrical single-cell recordings of responses to maximally exciting input (MEI) stimuli confirm chromatic selectivity of transient suppressed-by-contrast (tSbC) retinal ganglion cells (RGCs). (**a**) Spiking activity (top, raster plot; middle, estimated firing rate) of an OND RGC in response to different MEI stimuli (black bar indicates MEI stimulus duration; grey rectangle marks optimisation time window, from 1 to 1.66 s). Bottom: z-scored activity as a function of MEI stimulus, averaged across cells (solid circles w/ left y-axis, from Ca$^{2+}$ imaging, *N*=11 cells; open circles w/ right y-axis, from electrical spike recordings, *N*=4). Error bars show SD across cells. Colours as in *Figure 4*. (**b**) Like (**a**) but for a sustained ON $\alpha$ cell (G$_{24}$; *N*=4 cells, both for electrical and Ca$^{2+}$ recordings). (**c**) Different ON delayed (OND/tSbC, G$_{28}$) RGC (green) dye-loaded by patch pipette after cell-attached electrophysiology recording (z-projection; *x–y* plane). (**d**) Cell from (c, green) as side projection (*x–z*), showing dendritic stratification pattern relative to choline-acetyltransferase (ChAT) amacrine cells (tdTomato, red) within the inner plexiform layer (IPL).

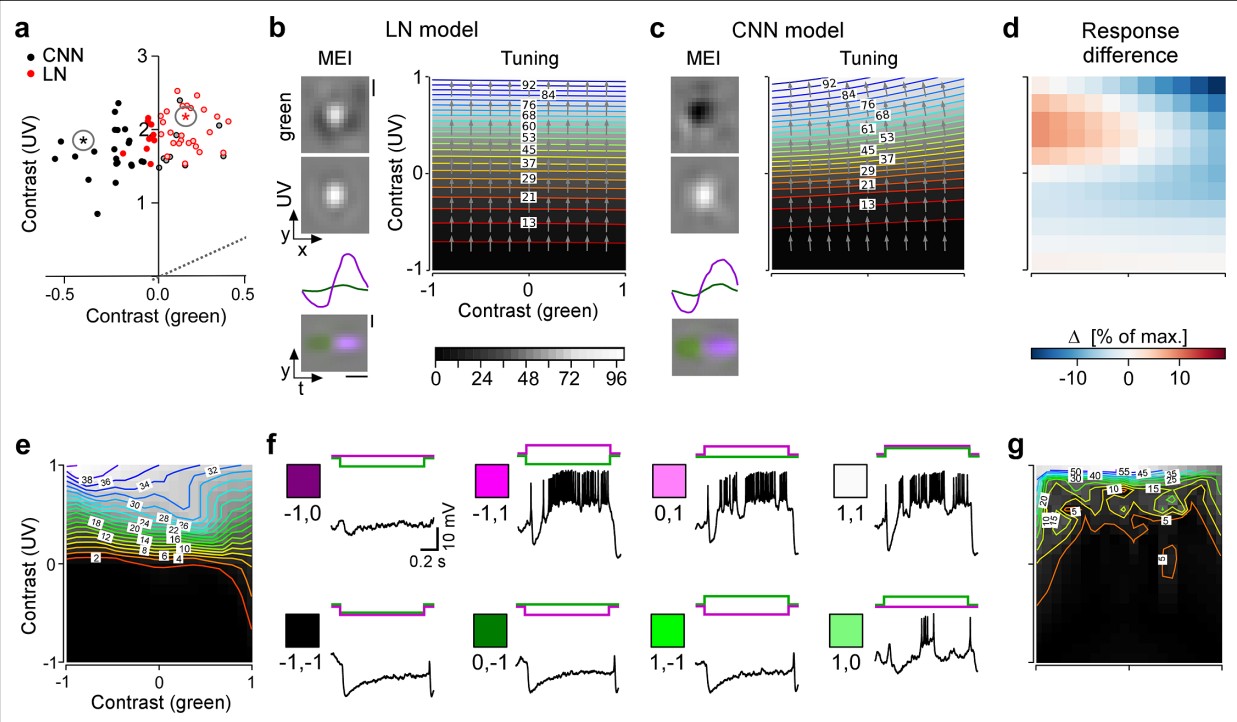

**Figure 6.** Chromatic contrast selectivity of $G_{28}$ retinal ganglion cells (RGCs) derives from a nonlinear transformation of stimulus space. (**a**) Distribution of green and UV maximally exciting input (MEI) centre contrast for a linear-nonlinear (LN) model (red) and a nonlinear convolutional neural network (CNN) model (black). Colour-opponent cells highlighted by filled marker. (**b, c**) Left: MEIs for an example cell of RGC group $G_{28}$, generated with the LN model (**b**) or the CNN model (**c**). The cell's MEI centre contrast for both models is marked in (**a**) by asterisks. Right: Respective tuning maps of example model neuron in chromatic contrast space. Contour colours and background greys represent responses in % of maximum response; arrows indicate the direction of the response gradient across chromatic contrast space. The tuning maps were generated by evaluating the model neurons on stimuli that were generated by modulating the contrast of the green (*x*-axis) and UV (*y*-axis) component of the MEI. In these plots, the original MEI is located at (–1, 1). More details in the Methods section. (**d**) Difference in response predicted between LN and CNN model (in % of maximum response). (**e**) Contour plot as in (**b, c**) but of activity vs. green and UV contrast for an example transient suppressed-by-contrast (tSbC) $G_{28}$ RGC measured in whole-cell current-clamp mode. Labels on the contour plot indicate spike count along isoresponse curves. (**f**) Traces are examples of responses at the 8 extremes of –100%, 0, or 100% contrast in each colour channel. Scale bars: (**b**), vertical 200 µm, horizontal 0.5 s; MEI scaling in (**c**) as in (**b**). (**g**) Same as (**e**) for a second example tSbC RGC.

The online version of this article includes the following figure supplement(s) for figure 6:

**Figure supplement 1.** Both LN and CNN model predict colour-opponency for a strongly colour-opponent G_{28} RGC.

To understand the nature of this dependency, we expanded the estimate of the model RGCs' tuning to colour contrast around the maximum (the MEI). We did this by mapping the model neurons' response and its gradient in 2D chromatic contrast space (***Figure 6b and c***). This analysis revealed that, indeed, $G_{28}$ RGCs have a nonlinear tuning for colour contrast: they are strongly UV-selective at lower contrasts, but become colour-opponent, i.e., additionally inhibited by green, for higher contrasts. For individual neurons with very strong colour opponency that extends over a large region of chromatic contrast space, also the LN model's approximation reflects this colour opponency, which demonstrates that the LN model can in principle model colour opponency, too (***Figure 6—figure supplement 1***).

We confirmed the model's predictions about $G_{28}$'s nonlinear tuning for colour contrast using electrical recordings as described above. The example $G_{28}$ (tSbC) cells shown in the figure exhibit similar nonlinear tuning in chromatic contrast space (***Figure 6e–g***). The first example cell's firing rate (***Figure 6f***) and, consequently, the tuning curve (***Figure 6e***) peak for UV[ON]-green[OFF] stimuli (top left in panel e; upper row, second from left in f), and are lower for UV[ON]-green[ON] stimuli (top right in panel e; upper row, far right in f), reflecting the suppressive effect of green contrast on the cell's response. The same is true for the second example cell (***Figure 6g***).

The nonlinearity in tuning to colour contrast of $G_{28}$ RGCs leads to a transformation of stimulus space (*Figure 6*) that amplifies the distance of colour-opponent stimuli from non-colour-opponent stimuli and is thereby expected to increase their discriminability. We therefore hypothesised that the representation of visual input formed by $G_{28}$ might serve to detect an ethologically relevant, colour-opponent feature from the visual scene.

## Visual context changes are characterised by changes in chromatic contrast

Chromatic contrast changes at the horizon (*Khani and Gollisch, 2021*; *Qiu et al., 2021*; *Gupta et al., 2022*; *Abballe and Asari, 2022*), and so does visual context: from sky to ground or vice versa. We therefore hypothesised that $G_{28}$ might leverage chromatic contrast as a proxy for detecting changes in visual context, such as might be caused by head or eye movements that cross the horizon. Detecting these changes in visual context could provide information that is crucial for interpreting signals in other RGC channels.

In our natural movie stimulus, the transitions between movie clips (*inter-clip transitions*; *Figure 1b*) can be categorised into those with and without change in visual context: ground-to-sky and sky-to-ground transitions for vertical movements with a change in visual context, and as controls, ground-to-ground and sky-to-sky transitions for horizontal movements without change in visual context. We calculated the contrast of these transitions in the green and UV channel to map them to chromatic contrast stimulus space (*Figure 7a*). We found that ground-to-ground and sky-to-sky transitions were distributed along the diagonal, indicating that they reflect largely achromatic contrast changes. The two transitions resembling visual input elicited by vertical movements crossing the horizon fell into the two colour-opponent quadrants of the stimulus space: sky-to-ground transitions in the lower right quadrant, and ground-to-sky transitions in the upper left quadrant (*Figure 7a and b*). The UV$^{ON}$-green$^{OFF}$ MEIs 28, the preferred stimuli of $G_{28}$, shared a location in stimulus space with ground-to-sky transitions, indicating that these two stimuli are similar in terms of chromatic contrast (*Figure 3i*).

## Chromatic contrast selectivity allows detecting visual context changes

Next, we tested if $G_{28}$ RGCs indeed respond strongly to visual context changes as occur in ground-to-sky transitions. To this end, we extracted the RGC responses to the inter-clip transitions, mapping their tuning to chromatic contrasts (*Figure 7—figure supplement 1*, *Figure 7—figure supplement 2*), and then averaged the resulting single-cell tuning maps for each RGC group (e.g. see *Figure 7c–e*). $G_{28}$ is most strongly tuned to full-field transitions in the upper left quadrant containing mostly ground-to-sky inter-clip transitions (*Figure 7c*) – unlike, for example, non-colour-opponent reference RGC groups from the slow ON and OFF response regime (*Figure 7d and e*).

Could a downstream visual area detect ground-to-sky visual context changes based on input from $G_{28}$ RGCs? To answer this question, we performed a linear signal detection analysis for each RGC by sliding a threshold across its responses to the inter-clip transitions, classifying all transitions that elicited an above-threshold response as ground-to-sky, and evaluating false-positive and true-positive rates (FPR and TPR, respectively) for each threshold (*Figure 7f*). Plotting the resulting TPRs for all thresholds as a function of FPRs yields a receiver operating characteristic (ROC) curve (*Fawcett, 2006*; *Figure 7f*, middle). The area under this curve (AUC) can be used as a measure of detection performance: it is equivalent to the probability that a given RGC will respond more strongly to a ground-to-sky transition than to any other type of transition. Indeed, $G_{28}$ RGCs achieved the highest AUC on average (*Figure 7f*, bottom, and *Figure 7g*; $G_{28}$, mean ± SD AUC ($N$=78 cells): 0.68 ± 0.08; two-sample permutation test $G_{28}$ vs. all other groups with at least $N$=4 cells (see Methods), significant for each group, with $\alpha = 0.0017$ Bonferroni-corrected for 30 multiple comparisons).

Ground-to-sky transitions, and therefore visual context changes, can also appear in the lower visual field, which is processed by the dorsal retina, where RGCs receive weaker UV input (*Szatko et al., 2020*). We recorded additional fields in the dorsal retina (*Figure 7—figure supplement 2*) and also found here that $G_{28}$ (tSbC) RGCs displayed the strongest tuning to ground-to-sky transitions among all dorsal RGCs (*Figure 7—figure supplement 3c-h*, for statistics, see legends).

Visual context changes triggered by different behaviours, such as locomotion and head or eye movements, will differ strongly with respect to their statistics – in particular with respect to their speed. Therefore, for $G_{28}$ (tSbC) RGCs to play a role in detecting context changes, their detection

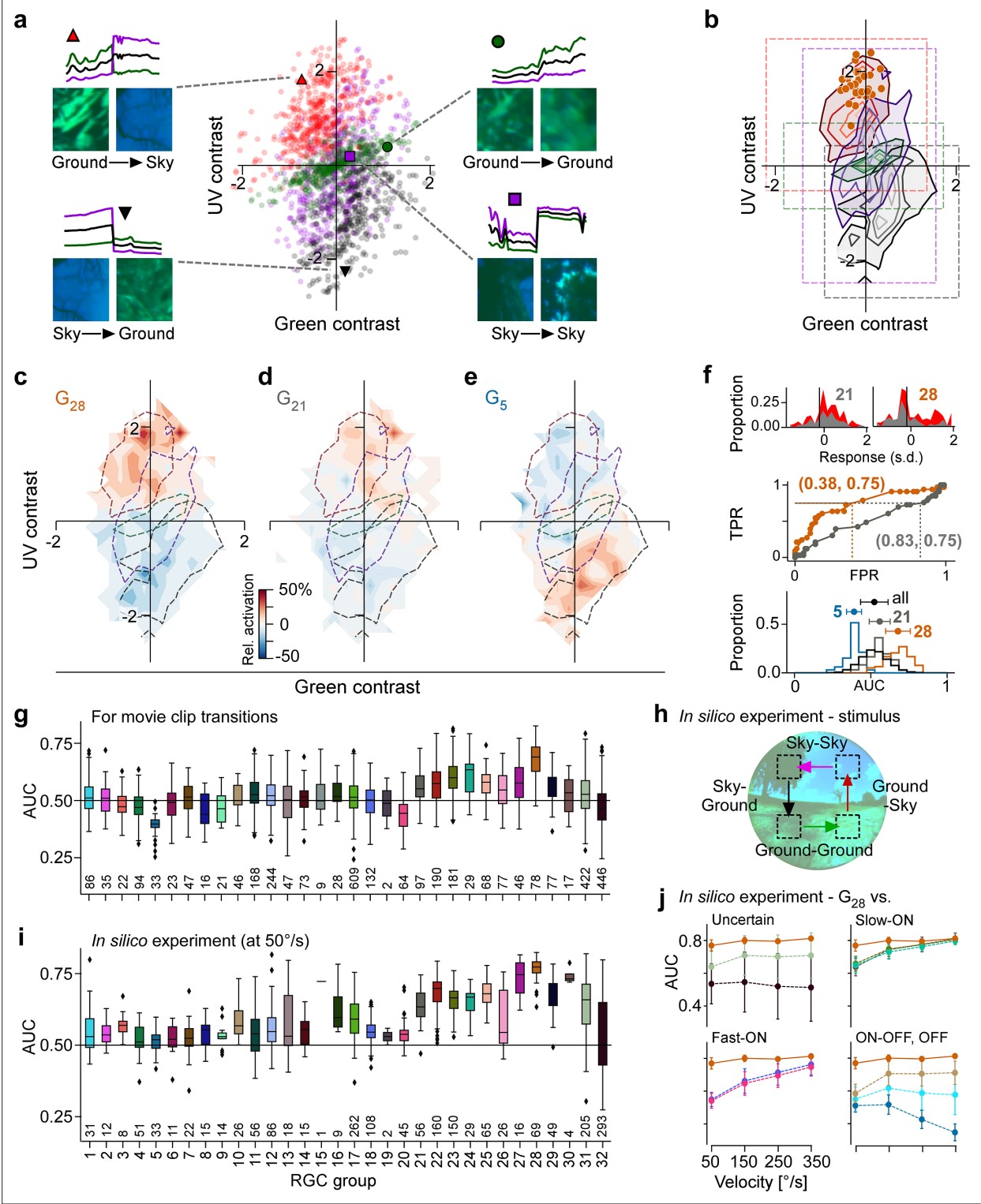

**Figure 7.** Chromatic contrast tuning allows detection of ground-to-sky transitions. (**a**) Distribution of green and UV contrasts of all movie inter-clip transitions (centre), separately for the four transition types, for each of which an example is shown: ground-to-sky (*N*=525, top left, red triangle), ground-to-ground (*N*=494, top right, green disk), sky-to-ground (*N*=480, bottom left, black downward triangle), and sky-to-sky (*N*=499, bottom right, purple square). Images show last and first frame of pre- and post-transition clip, respectively. Traces show mean full-field luminance of green and UV channels in last and first 1 s of pre- and post-transition clip. Black trace shows luminance averaged across colour channels. (**b**) Distributions as in (**a**), but shown as contours indicating isodensity lines of inter-clip transitions in chromatic contrast space. Density of inter-clip transitions was estimated separately for each

*Figure 7 continued on next page*

*Figure 7 continued*

type of transition from histograms within 10 × 10 bins that were equally spaced within the coloured boxes. Four levels of isodensity for each transition type shown, with density levels at 20% (outermost contour, strongest saturation), 40%, 60%, and 80% (innermost contour, weakest saturation) of the maximum density observed per transition: 28 sky-to-ground (black), 75 ground-to-ground (green), 42 sky-to-sky (purple), and 45 ground-to-sky (red) transitions per bin. Orange markers indicate locations of $N$=36 $G_{28}$ maximally exciting inputs (MEIs) in chromatic contrast space (*Figure 3i*). (**c**) Tuning map of $G_{28}$ retinal ganglion cells (RGCs) ($N$=78), created by averaging the tuning maps of the individual RGCs, overlaid with outermost contour lines from (**b**) (*Figure 7—figure supplement 2*). (**d, e**) Same as (**c**) for $G_{21}$ ((**g**), $N$=97) and $G_5$ ((**h**), $N$=33). (**f**) Top: Illustration of receiver operating characteristic (ROC) analysis for two RGCs, a $G_{21}$ (left) and a $G_{28}$ (right). For each RGC, responses to all inter-clip transitions were binned, separately for ground-to-sky (red) and all other transitions (grey). Middle: Sliding a threshold $d$ across the response range, classifying all transitions with response $> d$ as ground-to-sky, and registering the false-positive rate (FPR) and true-positive rate (TPR) for each threshold yields an ROC curve. Numbers in brackets indicate (FPR, TPR) at the threshold indicated by vertical line in histograms. Bottom: Performance for each cell, quantified as area under the ROC curve (AUC), plotted as distribution across AUC values for all cells (black), $G_{21}$ (grey), $G_5$ (blue), and $G_{28}$ (orange); AUC mean ± SD indicated as dots and horizontal lines above histograms. (**g**) Boxplot of AUC distributions per cell type. Boxes extend from first quartile ($Q_1$) to third quartile ($Q_3$) of the data; line within a box indicates median, whiskers extend to the most extreme points still within [$Q_1 - 1.5 \times \mathrm{IQR}$, $Q_3 + 1.5 \times \mathrm{IQR}$], IQR = inter-quartile range. Diamonds indicate points outside this range. All plot elements (upper and lower boundaries of the box, median line, whiskers, diamonds) correspond to actual observations in the data. Numbers of RGCs for each type are indicated in the plot. (**h**) Illustration of stimulus with transitions with (Sky-Ground, Ground-Sky) and without (Sky-Sky, Ground-Ground) context change at different velocities (50, 150, 250, and 350 °/s) used in *in silico* experiments in (**i, j**). (**i**) Like (**g**) but for model cells and stimuli illustrated in (**h**) at 50/s (see (**h**)). (**j**) AUC as function of transition velocity for $G_{28}$ (orange) vs. example RGC groups ('Uncertain', $G_{31,32}$; Slow-ON, $G_{21,23,24}$; Fast-ON, $G_{18,20}$; ON-OFF, $G_{10}$; OFF, $G_{1,5}$).

The online version of this article includes the following figure supplement(s) for figure 7:

**Figure supplement 1.** Example response traces to inter-clip transitions with and without context changes.

**Figure supplement 2.** Chromatic contrast tuning in the dorsal retina allows detection of ground-to-sky transitions.

**Figure supplement 3.** Simulations predict tSbC cells robustly detect context changes across different speeds.

performance should be robust across velocities. To test whether this is the case, we conducted additional in silico experiments where we ran model inference on stimuli that simulate transitions across the visual field with and without context change (*Figure 7h*) at different velocities: 50, 150, 250, and 350 visual degrees per second (° s⁻¹; see Methods; *Figure 7—figure supplement 3a and b*) The slowest speed simulated visual input as could be elicited by locomotion, and the fastest speed approached that of saccades (*Meyer et al., 2020*). We then performed an ROC analysis on the model cell responses. It should be noted that, because model predictions are noise-free, results from the ROC analysis based on simulated responses will overestimate detection performance. However, under the assumption of approximately equal noise levels across RGC groups, we can still draw conclusions about the relative performance of different RGC groups. This analysis confirmed that $G_{28}$ RGCs could distinguish ground-to-sky context changes from all other types of transitions robustly across different speeds (*Figure 7i and j*). Interestingly, the advantage of $G_{28}$ over other RGC groups in performing this detection task diminished with increasing speed (*Figure 7—figure supplement 3c and d*; see also Discussion).

Together, these analyses demonstrate that a downstream area, reading out from a single RGC group, would achieve the best performance in detecting ground-to-sky context changes if it based its decisions on inputs from $G_{28}$ RGCs, robustly across different lighting conditions (transitions between movie snippets), retinal location (ventral and dorsal), and speeds.

## Discussion

We combined large-scale recordings of RGC responses to natural movie stimulation with CNN-based modelling to investigate colour processing in the mouse retina. By searching the stimulus space *in silico* to identify *maximally exciting inputs* (MEIs), we found a novel type of chromatic tuning in tSbC RGCs. We revealed this RGC type's pronounced and unique selectivity for full-field changes from green-dominated to UV-dominated scenes, a stimulus that matches the chromatic statistics of ground-to-sky transitions in natural scenes. Therefore, we suggest that tSbC cells may signal context changes within their RF. Beyond our focus on tSbC cells, our study demonstrates the utility of an *in silico* approach for generating and testing hypotheses about the ethological relevance of sensory representations.

## Nonlinear approaches for characterising neuronal selectivities

We leveraged image-computable models in combination with an optimisation approach to search in dynamic, chromatic stimulus space for globally optimal inputs for RGCs, the MEIs. The resulting MEI represents the peak in the nonlinear loss landscape that describes the neuron's tuning in high-dimensional stimulus space. This approach has also been used to reveal the complexities and nonlinearities of neuronal tuning in monkey visual cortex area V4 (*Bashivan et al., 2019*; *Willeke et al., 2023*) and mouse area V1 (*Walker et al., 2019*; *Franke et al., 2022*).

Finding optimal stimuli via predictive models is by no means the only way to reveal nonlinear selectivity. Several alternative approaches exist (*Schwartz et al., 2006*; *Sharpee et al., 2004*; *Liu et al., 2017*; *Globerson et al., 2009*; *Maheswaranathan et al., 2023*) that could in principle also recover the type of tuning we report (although it is not a trivial question whether and under what conditions they would). More importantly, these approaches are not readily applicable to our data. Some approaches, such as spike-triggered covariance (*Schwartz et al., 2006*) or spike-triggered non-negative matrix factorisation (*Liu et al., 2017*), typically make assumptions about the distribution of input stimuli that are violated, or at least not guaranteed, for naturalistic stimuli, which consequently precludes using these methods to probe cells under their natural stimulus statistics (adaptations of these methods for naturalistic stimuli exist, e.g., see *Aljadeff et al., 2013*; *Aljadeff et al., 2016*). Other approaches, including maximally informative dimensions (*Sharpee et al., 2004*) or maximum noise entropy (*Globerson et al., 2009*), use information theory as a framework. They require estimating mutual information between stimulus and responses, which is a challenge when dealing with high-dimensional stimuli and continuous responses.

Predictive models, on the other hand, can handle high-dimensional input distributions well and can easily be adapted to different response modalities. However, one important limitation of our approach for identifying neuronal stimulus selectivities is that searching for the *maximally* exciting input will return a single input – even when there are several inputs that would elicit an equal response, such as ON and OFF stimuli for ON-OFF cells (see *Figure 3—figure supplement 1*, $G_{10}$ MEIs). A remedy for this limitation is to search for *diverse* exciting inputs by generating stimuli that are both highly effective at eliciting neuronal responses and at the same time distinct from one another. *Ding et al., 2023* used this approach to study bipartite invariance in mouse V1 (see also *Cadena et al., 2018*).

Another limitation is that identifying the MEI does not immediately provide insight into how the different stimulus dimensions contribute to the neuron's response, i.e., how varying the stimulus along these dimensions affects the neuron's response. However, differentiable models readily lend themselves to explore the interactions and contributions of different stimulus dimensions in generating the neuronal response, e.g., by inspecting the gradient field along dimensions of interest as done here, or by searching for locally optimal stimulus perturbations (*Goldin et al., 2022*). These models can also be used to understand better the properties that distinguish cell types from each other by generating *most discriminative stimuli*, as recently demonstrated for RGCs in mouse and marmoset (*Burg et al., 2023*).

Together, these studies showcase the versatility of the toolkit of optimisation-based approaches at characterising nonlinear neuronal operations in high-dimensional, natural stimulus spaces. We add to this toolkit by first searching for a globally optimal stimulus, and then searching locally in its vicinity to map the cells' loss landscape around the peak.

## Circuit mechanisms for colour opponency in tSbC RGCs

Most previous studies of colour opponency in the mouse retina have identified sparse populations of colour-opponent RGCs that have not been systematically assigned to a particular functional type (*Szatko et al., 2020*; *Khani and Gollisch, 2021*; *Gouras and Ekesten, 2004*). The only studies that have examined the mechanisms of colour opponency in identified mouse RGC types showed a centre-surround organisation, with RF centre and surround having different chromatic preferences (*Chang et al., 2013*; *Joesch and Meister, 2016*; *Stabio et al., 2018*, but see *Sonoda et al., 2020*). While we do not specifically analyse centre-surround opponency in this study, we see a similar trend as described previously in many RGC types, with stronger surrounds in the green channel relative to the UV channel (see *Figure 4a*, *Figure 3—figure supplement 1*). tSbC RGCs, in contrast, respond to spatially co-extensive colour-opponent stimuli, functionally reminiscent of colour-opponent RGCs in guinea pig (*Yin et al., 2009*) and ground squirrels (*Michael, 1968*).

In mice, centre-surround opponency has been attributed to the opsin gradient (*Chang et al., 2013*) and rod contributions in the outer retina (*Joesch and Meister, 2016*; *Szatko et al., 2020*), whereas the circuitry for spatially co-extensive opponency remains unknown. It seems unlikely, though, that the opsin gradient plays a major role in the tSbC cell's colour opponency, because both ventral and dorsal tSbC cells preferentially responded to full-field green-to-UV transitions. In primates, spatially co-extensive colour opponency in small bistratified RGCs is thought to arise from the selective wiring of $S^{ON}$ and $M/L^{OFF}$ bipolar cells onto the inner and outer dendritic strata, respectively (*Dacey and Lee, 1994*, but see *Field et al., 2007*). A similar wiring pattern seems unlikely for tSbC RGCs, since their inner dendrites do not co-stratify with the S-ON (type 9) bipolar cells, nor do their outer dendrites co-stratify with the candidate M-OFF bipolar cell (type 1) (*Behrens et al., 2016*). The bistratified dendritic arbour distinguishes the mouse tSbC also from the colour-opponent ON RGC type in guinea pig, which is monostratified (*Yin et al., 2009*).

The large RF centres of the tSbC cells, extending well beyond their dendritic fields, come from a non-canonical circuit, in which tonic inhibition onto the RGC via $GABA_B$ receptors is relieved via serial inhibition from different amacrine cells using $GABA_C$ receptors (*Mani and Schwartz, 2017*). An intriguing possibility is that a colour-selective amacrine cell is part of this circuit, perhaps supporting chromatically tuned disinhibition in the absence of selective wiring from the aforementioned cone-selective bipolar cells onto the RGC.

## A new functional role for tSbC RGCs

Suppressed-by-contrast responses have been recorded along the early visual pathway in dorsal lateral geniculate nucleus (dLGN), superior colliculus (SC), and primary visual cortex (V1) (*Niell and Stryker, 2010*; *Piscopo et al., 2013*; *Ito et al., 2017*), with their function still being debated (*Masland and Martin, 2007*). In the retina, three types of SbC RGCs have so far been identified (reviewed in *Jacoby and Schwartz, 2018*), among them the tSbC cell (*Mani and Schwartz, 2017*; *Tien et al., 2015*; *Tien et al., 2022*). Despite their relatively recent discovery, tSbC RGCs have been suggested to play a role in several different visual computations. The first report of their light responses in mice connected them to the SbC RGCs previously discovered in rabbit, cat, and macaque, and suggested a role in signalling self-generated stimuli, perhaps for saccade suppression (*Tien et al., 2015*). Aided by a new intersectional transgenic line to selectively label tSbC RGCs (*Tien et al., 2022*), their projections were traced to areas in SC, v- and dLGN, and nucleus of the optic tract (NOT). The latter stabilises horizontal eye movements; however, as the medial terminal nucleus, which serves stabilisation of vertical eye movements, lacks tSbC innervation, it is unclear whether and how these RGCs contribute to gaze stabilisation.

A retinal study identified the circuit mechanisms responsible for some of the unique spatial and temporal response properties of tSbC cells and suggested a possible role in defocus detection to drive emmetropization in growing eyes and accommodation in adults (*Mani and Schwartz, 2017*; *Baden et al., 2017*). Here, we identified another potential role for these RGCs in vision based on the chromatic properties of their RFs: signalling visual context changes (see next section). These different possible functional roles are not mutually exclusive, and might even be complementary in some cases, highlighting the difficulty in assigning single features to distinct RGC types (*Schwartz and Swygart, 2020*). In particular, the centre colour opponency that we discovered in tSbC RGCs could serve to enhance their role in defocus detection by adding a directional signal (myopic vs. hyperopic) based on the chromatic aberration of lens and cornea (*Gawne and Norton, 2020*). Future studies may test these theories by manipulating these cells in vivo using the new transgenic tSbC mouse line (*Tien et al., 2022*).

## Behavioural relevance of context change detection

The horizon is a prominent landmark in visual space: it bisects the visual field into two regions, ground and sky. This is particularly relevant in animals like mice, where eye motion largely accounts for head movements and keeps the visual field stable with respect to the horizon (*Meyer et al., 2020*). Visual stimuli carry different meaning depending on where they occur relative to the horizon, and context-specific processing of visual inputs is necessary for selecting appropriate behavioural responses (reviewed in *Evans et al., 2019*). For example, it is sensible to assume that a looming stimulus above the horizon is a predator, the appropriate response to which would be avoidance (i.e. escape or

freezing). A similar stimulus below the horizon, however, is more likely to be harmless or even prey. To allow for time-critical perceptual decisions – predator or prey – and corresponding behavioural response selection – avoidance or approach – it might be useful for stimulus information (e.g. dark moving spot) and contextual information to converge early in the visual circuitry.

Notably, VGluT3-expressing amacrine cells (a 'hub' for distributing information about motion) represent a shared element in upstream circuitry, providing opposite-sign input to tSbC and to RGCs implicated in triggering avoidance behaviour, such as tOFF $\alpha$ (*Krieger et al., 2017*; *Münch et al., 2009*) and W3 cells (*Zhang et al., 2012*). In downstream circuitry, SbC inputs have been found to converge with 'conventional' RGC inputs onto targets in dLGN and NOT; whether tSbC axons specifically converge with tOFF $\alpha$ or W3 axons remains to be tested. Such convergence may allow 'flagging' the activity of these RGCs with their local context (sky/threat or ground/no threat).

Depending on the behaviour that elicits a context change – be it a head or eye movement or locomotion – the parameters of the incoming stimulus, such as illumination level and velocity, may change. To be behaviourally useful, a context-change-flagging signal needs to be reliable and robust across these different stimulus parameters. While many slow-ON RGCs achieve high detection performance at higher transition velocities, probably reacting to the increasingly flash-like stimuli, tSbC/$G_{28}$ RGCs were the only type with robustly high performance across different levels of illumination and all simulated speeds.

## In silico approaches to linking neural tuning and function

Modelling studies have advanced our understanding of the complexity and organisation of retinal processing in recent years. It is helpful to consider the contributions of different studies in terms of three perspectives on the retinal encoding of natural scenes: the circuit perspective ('how?'), the normative perspective ('why?'), and the coding perspective ('what?') (*Marr, 2010*; *Karamanlis et al., 2022*). For example, an *in silico* dissection of a CNN model of the retina offered explanations on how the surprisingly complex retinal computations, such as motion reversal, omitted stimulus response, and polarity reversal, emerge from simpler computations within retinal circuits (*Maheswaranathan et al., 2023*; *Tanaka et al., 2019*). From the normative perspective, networks trained on an efficient coding objective accurately predicted the coordination of retinal mosaics (*Roy et al., 2021*).

Here, we proposed an approach that allows investigating the complexity of retinal processing simultaneously from the coding and the normative perspectives: A global search for most exciting mouse RGC inputs in dynamic, chromatic stimulus space provides answers to the question of *what* it is that retinal neurons encode. Interpreting the abstract features extracted by the retina against the backdrop of natural stimulus space points to *why* these features might be behaviourally relevant. And finally, classifying individual RGCs into types then allows to bring in the circuit perspective through targeted experiments aimed at dissecting *how* specific retinal computations are implemented.

## Methods
### Animals and tissue preparation

All imaging experiments were conducted at the University of Tübingen; the corresponding animal procedures were approved by the governmental review board (Regierungspräsidium Tübingen, Baden-Württemberg, Konrad-Adenauer-Str. 20, 72072 Tübingen, Germany) and performed according to the laws governing animal experimentation issued by the German Government. All electrophysiological experiments were conducted at Northwestern University; the corresponding animal procedures were performed according to standards provided by Northwestern University Center for Comparative Medicine and approved by the Institutional Animal Care and Use Committee (IACUC).

For all imaging experiments, we used 4- to 15-week-old C57Bl/6J mice (*N*=23; JAX 000664) of either sex (10 male, 13 female). These animals were housed under a standard 12 hr day/night rhythm at 22° and 55% humidity. On the day of the recording experiment, animals were dark-adapted for at least 1 hr, then anaesthetised with isoflurane (Baxter) and killed by cervical dislocation. All following procedures were carried out under very dim red (>650 nm) light. The eyes were enucleated and hemisected in carboxygenated (95% $O_2$, 5% $CO_2$) artificial cerebrospinal fluid (ACSF) solution containing (in mM): 125 NaCl, 2.5 KCl, 2 CaCl$_2$, 1 MgCl$_2$, 1.25 NaH$_2$PO$_4$, 26 NaHCO$_3$, 20 glucose, and 0.5 L-glutamine at pH 7.4. Next, the retinae were bulk-electroporated with the fluorescent Ca$^{2+}$

indicator Oregon-Green BAPTA-1 (OGB-1), as described earlier (*Briggman and Euler, 2011*). In brief, the dissected retina was flat-mounted onto an Anodisc (#13, 0.2 μm pore size, GE Healthcare) with the RGCs facing up, and placed between a pair of 4 mm horizontal plate electrodes (CUY700P4E/L, Nepagene/Xceltis). A 10 μl drop of 5 mM OGB-1 (hexapotassium salt; Life Technologies) in ACSF was suspended from the upper electrode and lowered onto the retina. Next, nine pulses (≈9.2 V, 100 ms pulse width, at 1 Hz) from a pulse generator/wide-band amplifier combination (TGP110 and WA301, Thurlby handar/Farnell) were applied. Finally, the tissue was placed into the microscope's recording chamber, where it was perfused with carboxygenated ACSF (at ≈36°C) and left to recover for ≥30 min before recordings started. To visualise vessels and damaged cells in the red fluorescence channel, the ACSF contained ≈0.1 μM Sulforhodamine-101 (SR101, Invitrogen) (*Euler et al., 2009*). All procedures were carried out under dim red (>650 nm) light.

For electrophysiology experiments, we used ChAT-Cre (JAX 006410) × Ai14 (JAX 007914) mice on a C57Bl/6J background (*N*=2, male, aged 27 and 30 weeks). Mice were housed with siblings in groups up to 4, fed normal mouse chow and maintained on a 12:12 hr light/dark cycle. Before the experiment, mice were dark-adapted overnight and sacrificed by cervical dislocation. Retinal tissue was isolated under infrared illumination (900 nm) with the aid of night-vision goggles and IR dissection scope attachments (BE Meyers). Retinal orientation was identified using scleral landmarks (*Wei et al., 2010*), and preserved using relieving cuts in cardinal directions, with the largest cut at the dorsal retina. Retinas were mounted on 12 mm poly-D-lysine-coated glass affixed to a recording dish with grease, with the GCL up. Oxygenation was maintained by superfusing the dish with carboxygenated Ames medium (US Biological, A1372-25) warmed to 32°C. For cell-attached single-cell recordings, we used Symphony software (*Cafaro et al., 2019*) with custom extensions (*Schwartz and Ala-Laurila, 2024*).

Owing to the exploratory nature of our study, we did not use randomisation and blinding. No statistical methods were used to predetermine sample size.

## Two-photon calcium imaging

We used a MOM-type two-photon microscope (designed by W Denk, purchased from Sutter Instruments) (*Euler et al., 2009*; *Euler et al., 2019*), which was equipped with a mode-locked Ti:sapphire laser (MaiTai-HP DeepSee, Newport Spectra-Physics) tuned to 927 nm, two fluorescence detection channels for OGB-1 (HQ 510/84, AHF/Chroma) and SR101 (HQ 630/60, AHF), and a water immersion objective (CF175 LWD x 16/0.8W, DIC N2, Nikon, Germany). Image acquisition was performed with custom-made software (ScanM by M Müller and TE) running under IGOR Pro 6.3 for Windows (Wavemetrics), taking time-lapsed 64 × 64 pixel image scans (≈ (100 μm)$^2$) at 7.8125 Hz (*Figure 1c*). For simplicity, we refer to such a time-lapsed scan of a local population of GCL cells as a 'recording'. Despite the low frame rate, the Ca$^{2+}$ responses can be related to the spike rate (*Trapani et al., 2023*; *Román Rosón et al., 2019*; *Berens et al., 2018*; *Theis et al., 2016*). For documenting the position of the recording fields, the retina under the microscope was oriented such that the most ventral edge pointed always towards the experimenter. In addition, higher resolution images (512 ×512 pixel) were acquired and recording field positions relative to the optic nerve were routinely logged.

## Data preprocessing

Ca$^{2+}$ traces were extracted for individual ROIs as described previously (*Baden et al., 2016*; *Szatko et al., 2020*). Extracted traces $\mathbf{c}_{raw}$ were then detrended to remove slow drifts in the recorded signal that were unrelated to changes in the neural response. First, a smoothed version of the traces, $\mathbf{c}_{smooth}$, was calculated by applying a Savitzky-Golay filter of third polynomial order and a window length of 60 s using the SciPy implementation `scipy.signal.savgol_filter`. This smoothed version was then subtracted from the raw traces to yield the detrended traces.

$$\mathbf{c}_{detrend} = \mathbf{c}_{raw} - \mathbf{c}_{smooth}$$

To make traces non-negative ($\mathbf{c}_+$), we then clipped all values smaller than the 2.5th percentile, $\eta_{2.5}$, to that value, and then subtracted $\eta_{2.5}$ from the detrended traces:

$$\mathbf{c}_+ = \mathbf{c}_{detrend} - \eta_{2.5}$$

This procedure (i.e. clipping to, and subtracting $\eta_{2.5}$) was more robust than simply subtracting the minimum.

Finally, traces were then divided by the standard deviation within the time window before stimulus start at $t_0$:

$$\mathbf{c} := \mathbf{c}_{final} = \frac{\mathbf{c}_{nn}}{SD(\mathbf{c}_{+[:t_0]})}$$

For training the model on movie response, we then estimated firing rates $\mathbf{r}$ from the detrended Ca$^{2+}$ traces $\mathbf{c}$ using the package C2S (https://github.com/lucastheis/c2s, copy archived at *lucastheis, 2016*; *Theis et al., 2016*).

## Inclusion criteria

We applied a sequence of quality filtering steps to recorded cells before analysis illustrated in *Figure 1—figure supplement 1*. As a first step, we applied a general response quality criterion, defined as a sufficiently reliable response to the MB stimulus (as quantified by a quality index $QI_{MB} > 0.6$), *or* a sufficiently reliable response to the chirp stimulus (as quantified by a quality index $QI_{chirp} > 0.35$). The quality index is defined as in *Baden et al., 2016*:

$$QI = \frac{\mathrm{Var}[\langle \boldsymbol{r} \rangle_i]_t}{\langle \mathrm{Var}[\boldsymbol{r}]_t \rangle_i}$$

where $\boldsymbol{r}$ is the $T$ by $I$ response matrix (time samples by stimulus repetitions) and $\langle \rangle_x$ and $\mathrm{Var}[]_x$ denote the mean and variance across the indicated dimension $x$, respectively.

The second and third step made sure only cells were included that were assigned to a ganglion cell group (i.e. group index between 1 and 32) with sufficient confidence. Confidence is defined as the probability assigned to the predicted class by the random forest classifier (see *Qiu et al., 2023*), and the threshold was set at $\geq 0.25$.

The fourth step made sure only cells with a sufficient model prediction performance, defined as an average single-trial test set correlation of $\langle C(\hat{r}^{(n)}, r_i^{(n)}) \rangle_i > 0.3$, were included.

All cells passing steps 1–3 were included in the horizon detection analysis (*Figure 7*); all cells passing steps 1–4 were included in the MEI analysis (*Figure 3*); the 'red' cells passing steps 1–4 were included in the MEI validation analysis (*Figure 4*). In the process of analysing MEIs, we fitted DoGs to their green and UV spatial component (see Methods section Concentric anisotropic 2D DoG fit). For the analysis of MEI properties (temporal frequency, centre size, chromatic contrast), we only included cells with a sufficient DoG goodness-of-fit, determined as a value of the cost function of <0.11 for both green and UV on the resulting DoG fit. This threshold was determined by visual inspection of the DoG fits and led to the inclusion of 1613 out of 1947 RGCs in the MEI property analysis.

## Visual stimulation

For light stimulation (imaging experiments), we projected the image generated by a digital light processing projector (lightcrafter DPM-FE4500MKIIF, EKB Technologies Ltd) through the objective onto the tissue. The lightcrafter featured a light-guide port to couple in external, band-pass-filtered UV and green LEDs (light-emitting diodes) (green: 576 BP 10, F37-576; UV: 387 BP 11, F39-387; both AHF/Chroma) (*Franke et al., 2019*). To optimise spectral separation of mouse M- and S-opsins, LEDs were band-pass-filtered (390/576 dual-band, F59-003, AHF/Chroma). LEDs were synchronised with the microscope's scan retrace. Stimulator intensity (as photoisomerisation rate, $10^3$ $P^* s^{-1}$ per cone) was calibrated to range from $\approx 0.5$ (black image) to $\approx 20$ for M- and S-opsins, respectively. Additionally, we estimated a steady illumination component of $\approx 10^4$ $P^* s^{-1}$ per cone to be present during the recordings because of two-photon excitation of photopigments (*Euler et al., 2009*; *Euler et al., 2019*). Before data acquisition, the retina was adapted to the light stimulation by presenting a binary noise stimulus (20 × 15 matrix, (40 µm)$^2$ pixels, balanced random sequence) at 5 Hz for 5 min to the tissue. Stimuli were presented using the software RRID:SCR_016985, QDSpy (https://github.com/eulerlab/QDSpy).

For electrophysiology experiments, stimuli were presented using a digital projector (DPM-FE4500MKII, EKB Technologies Ltd) at a frame rate of 60 Hz and a spatial resolution of 1140 × 912 pixels (1.3 µm per pixel) focused on the photoreceptor layer. Neutral density filters (Thorlabs),

a triple-band pass filter (405 BP 20, 485 BP 20, 552 BP 16; 69,000×, Chroma), and a custom LED controller circuit were used to attenuate the light intensity of stimuli either to match that of the $Ca^{2+}$ imaging experiments (for MEI presentation) or to range from ≈0 to 200 $P^*s^{-1}$ per rod (for cell identification). Stimuli were presented using Symphony software (https://symphony-das.github.io/) with custom extensions (https://github.com/Schwartz-AlaLaurila-Labs/sa-labs-extension, copy archived at *Schwartz-AlaLaurila-Labs, 2024*).

### Identifying RGC types

To functionally identify RGC groups in the $Ca^{2+}$ imaging experiments, we used our default 'fingerprinting' stimuli, as described earlier (*Baden et al., 2016*). These stimuli included a full-field (700 µm in diameter) chirp stimulus, and a 300 ×1,000 µm bright bar moving at 1000 µm/s in eight directions across the recording field (with the shorter edge leading; *Figure 1b*).

The procedure and rationale for identifying cells in the electrophysiological recordings is presented in *Goetz et al., 2022*. Cells with responses that qualitatively matched that of the OND and ON $\alpha$ types were included in the study. Following recording, cells were filled with Alexa Fluor-488 by patch pipette and imaged under a two-photon microscope. Dendrites were traced in Fiji (NIH) using the SNT plugin (*Arshadi et al., 2021*). Dendritic arbours were computationally flattened using a custom MATLAB tool (https://doi.org/10.5281/zenodo.6578530) based on the method in *Sümbül et al., 2014*, to further confirm their identity as morphological type 73 from *Bae et al., 2018*.

### Mouse natural movies

The natural movie stimulus consisted of clips of natural scenes recording outside in the field with a specialised, calibrated camera (*Qiu et al., 2021*). This camera featured a fish-eye lens, and two spectral channels, UV (band-pass filter F37-424, AHF, > 90% transmission at 350–419 nm) and green (F47-510, >90%, 470–550 nm, AHF), approximating the spectral sensitivities of mouse opsins (*Jacobs et al., 2004*). In mice, eye movements often serve to stabilise the image on the retina during head movements (*Meyer et al., 2020*). Therefore, the camera was also stabilised by mounting it on a gimbal. As a result, the horizon bisected the camera's visual field.

A *mouse cam movie* frame contained a circular field of view of 180° corresponding to 437 pixels along the diameter. To minimise the influence of potential chromatic and spatial aberrations introduced by the lenses, we focused on image cut-outs (crops; 30 × 26, equivalent to 72 × 64 pixels in size) from upper and lower visual field, centred at [28, 56] and [-42, –31], respectively, relative to the horizon (for details, see *Qiu et al., 2021*). Our *stimulus movie* consisted of 113 movie clips, each 150 frames (= 5 s) long. 108 clips were randomly reordered for each recording and split into two 54 clips-long *training sequences*. The remaining 5 clips formed a fixed *test sequence* that was presented before, in between, and after the training sequences (*Figure 1b*). To keep intensity changes at clip transitions small, we only used clips with mean intensities between 0.04 and 0.22 (for intensities in [0, 1]). For display during the experiments, intensities were then mapped to the range covered by the stimulator, i.e., [0, 255].

### CNN model of the retina

We trained a CNN model to predict responses of RGCs to a dichromatic natural movie. The CNN model consisted of two modules, a convolutional core that was shared between all neurons, and a readout that was specific for each neuron (*Klindt et al., 2017*).

The core module was modelled as a two-layer CNN with 16 feature channels in each layer. Both layers consisted of space-time separable 3D convolutional kernels followed by a batch normalisation layer and an ELU (exponential linear unit) nonlinearity. In the first layer, sixteen 2 × 11 × 11 × 21 ($c$=#input channels (green and UV) × $h$=height ×$w$=width × $t$=#frames) kernels were applied as valid convolution; in the second layer, sixteen 16 × 5 × 5 × 11 kernels were applied with zero padding along the spatial dimensions. We parameterised the temporal kernels as Fourier series and added one time stretching parameter per recording to account for inter-experimental variability affecting the speed of retinal processing. More precisely, every temporal kernel was represented by the first $k$ sine and cosine functions, with trainable weights and phases, on an evenly spaced temporal grid, where $k = 7$ for the first layer, and $k = 3$ for the second layer. Additionally, we introduced a trainable stretch parameter for every recording to account for faster and slower response kernels. For example, the

first layer temporal kernels are 21 steps long. Then, in order to stay well under the Nyquist limit, we parameterise the kernels with $k = 21/3 = 7$ sines and cosines.

For each of those sines and cosines a weight $(\alpha, \beta)$ is learned to represent the shape of the temporal responses kernel (shared among cells within a recording). Per scan $i$, the time grid $t$ (21 steps from 0 to 1) is stretched by a factor $\tau_i$ to account for different response speeds. To avoid adding additional cycles (e.g. for stretch factors $\tau > 1$) this is masked by an exponential envelope

$$\epsilon(\tau) = \frac{1}{1 + \exp -(t + \frac{21 \cdot 0.95}{\tau})} \tag{1}$$

Thus,

$$w_i = \sum_{j}^{k} \alpha_j \sin(2\pi \cdot \tau_i \cdot t \cdot \epsilon(\tau_i)) + \beta_j \cos(2\pi \cdot \tau_i \cdot t \cdot \epsilon(\tau_i)). \tag{2}$$

is the temporal kernel parameterisation, that allows the model to learn a shared temporal filter that is made faster or slower for each specific scan (*Zhao et al., 2020*).

In the readout, we modelled each cell's spatial receptive field (RF) as a 2D isotropic Gaussian, parameterised as $\mathcal{N}(\mu_x, \mu_y; \sigma)$. We then modelled the neural response as an affine function of the core feature maps weighted by the spatial RF, followed by a softplus nonlinearity.

For the linearised version of the model, the architecture was exactly the same except for the fact that there was no ELU nonlinearity after both convolutional layers. The resulting CNN was therefore equivalent to an LN model.

## Model training and evaluation

We trained our network by minimising the Poisson loss

$$\sum_{n=1}^{N} \left( \hat{\boldsymbol{r}}^{(n)} - \boldsymbol{r}^{(n)} \log \hat{\boldsymbol{r}}^{(n)} \right)$$

where $N$ is the number of neurons, $\boldsymbol{r}^{(n)}$ is the measured and $\hat{\boldsymbol{r}}^{(n)}$ the predicted firing rate of neuron $n$ for an input of duration $t$=50 frames. We followed the training schedule of *Lurz et al., 2021*. Specifically, we used early stopping (*Prechelt, 1998*) on the correlation between predicted and measured neuronal responses on the validation set, which consisted of 15 out of the 108 movie clips. If the correlation failed to increase during any five consecutive passes through the entire training set (epochs), we stopped the training and restored the model to the best performing model over the course of training. We went through four cycles of early stopping, restoring the model to the best performing, and continuing training, each time reducing the initial learning rate of 0.01 by a learning rate decay factor of 0.3. Network parameters were iteratively optimised via stochastic gradient descent (SGD) using the Adam optimiser (*Kingma and Ba, 2015*) with a batch size of 32 and a chunk size (number of frames for each element in the batch) of 50. For all analyses and MEI generation, we used an ensemble of models as described in *Franke et al., 2022*. Briefly, we trained five instances of the same model initialised with different random seeds. Inputs to the ensemble model were passed to each member and the final ensemble model prediction was obtained by averaging the outputs of the five members. For ease of notation, we thus redefine $\hat{\boldsymbol{r}}^{(n)}$ to be the *ensemble* model prediction.

After training, we evaluated model performance for each modelled neuron $n$ as the correlation to the mean, i.e., the correlation between predicted response $\hat{r}^{(n)}$ and measured response $r^{(n)}$ to the held-out test sequence, the latter averaged across three repetitions $i = \{1, 2, 3\}$: $C\left( \hat{r}^{(n)}, \left\langle r_i^{(n)} \right\rangle_i \right)$. Unlike the single-trial correlation $C(\hat{r}^{(n)}, r_i^{(n)})$ which is always limited to values $< 1$ by inherent neuronal noise, a perfect model can in theory achieve a value of 1 for the correlation to the mean, in the limit of infinitely many repetitions when the sample average $\langle r^{(n)_i} \rangle_i$ is a perfect estimate of the true underlying response $\rho^{(n)}$. The observed correlation to the mean can thus be interpreted as an estimate of the fraction of the maximally achievable correlation achieved by our model. For deciding which cells to exclude from analysis, we used average single-trial correlation $(\langle C(\hat{r}^{(n)}, r_i^{(n)}) \rangle_i)$ since this measure

reflects both model performance as well as reliability of the neuronal response to the movie stimulus for neuron $n$ (see also Methods section on Inclusion criteria).

## Synthesising MEIs

We synthesised MEIs for RGCs as described previously (*Walker et al., 2019*). Formally, for each model neuron $n$ we wanted to find

$$\boldsymbol{x}^{*(n)} = \arg \max_{\boldsymbol{x}} \langle \hat{\boldsymbol{r}}^{(n)}(\boldsymbol{x})_{30:50} \rangle_t, \tag{3}$$

i.e., the input $\boldsymbol{x}^{*(n)}$ where the model neuron's response $\langle \hat{r}(\boldsymbol{x})_{30:50} \rangle_t$, averaged across frames 30–50, attains a maximum, subject to norm and range constraints (see below). To this end, we randomly initialised an input $\boldsymbol{x}_0^{(n)} \in \mathcal{R}^{c \times w \times h \times t}$ of duration $t$=50 frames with Gaussian white noise, and then iteratively updated $x_i^{(n)}$ according to the gradient of the model neuron's response:

$$\boldsymbol{x}_{i+1}^{(n)} = \boldsymbol{x}_i^{(n)} + \lambda \frac{\delta}{\delta \boldsymbol{x}_i^{(n)}} \langle \hat{\boldsymbol{r}}^{(n)}(\boldsymbol{x}_i^{(n)})_{30:50} \rangle_t, \tag{4}$$

where $\lambda = 10$ was the learning rate. The optimisation was performed using SGD, and was subject to a norm and a range constraint. The norm constraint was applied jointly across both channels and ensured that the L2 norm of each MEI did not exceed a fixed budget $b$ of 30. The norm-constrained MEI $\tilde{\boldsymbol{x}}_i^{(n)}$ was calculated at each iteration as

$$\tilde{\boldsymbol{x}}_i^{(n)} = \frac{b}{\|\boldsymbol{x}_i^{(n)}\|_2} \times \boldsymbol{x}_i^{(n)} \tag{5}$$

The range constraint was defined and applied for each colour channel separately and ensured that the range of the MEI values stayed within the range covered by the training movie. This was achieved by clipping values of the MEI exceeding the range covered by the training movie to the minimum or maximum value. Optimisation was run for at least 100 iterations, and then stopped when the number of iterations reached 1000, or when it had converged (whichever occurred first). Convergence was defined as 10 consecutive iterations with a change in model neuron activation of less than 0.001; model neuron activations ranged from ≈1 to ≈10. We denote the resulting MEI for neuron $n$ as $\boldsymbol{x}^{*(n)}$.

## Analysing MEIs

We analysed MEIs to quantify their spatial, temporal, and chromatic properties.

### Spatial and temporal components of MEIs

For each colour channel $c$, we decomposed the spatiotemporal MEIs into a spatial component and a temporal component by singular value decomposition:

$$U, S, V = \mathrm{svd}(\boldsymbol{x}_c^{*(n)})$$

with $x_c^{*(n)} \in \mathcal{R}^{50 \times 288}$ for $c \in$ [green, UV] is the MEI of neuron $n$ in a given colour channel with its spatial dimension (18 ×16=288) flattened out. As a result, any spatiotemporal dependencies are removed and we only analyse spatial and temporal properties separately. The following procedures were carried out in the same manner for the green and the UV component of the MEI, and we drop the colour channel index $c$ for ease of notation. The temporal component is then defined as the first left singular vector, $U_{:1}$, and the spatial component is defined as the first right singular vector, $V_{:1}^T$, reshaped to the original dimensions 18 × 16 .

### Concentric anisotropic 2D DoG fit

We modelled the spatial component as concentric anisotropic DoG using the nonlinear least-squares solver `scipy.optimize.least_squares` with soft-L1 loss function (*Gupta et al., 2022*). The DoGs were parameterised by a location $(\mu_x, \mu_y)$ shared between centre and surround, amplitudes $A^c$, $A^s$, variances $(\sigma_x^c, \sigma_y^c)$, $(\sigma_x^s, \sigma_y^s)$, and rotation angles $\theta^c$, $\theta^s$ separately for centre and surround:

$$\text{DoG} = G^c - G^s$$

with

$$G^c(x, y) = \text{A}^c \exp(-f^c(x - \mu_x)^2$$
$$+2g^c(y - \mu_y)(x - \mu_x)$$
$$+h^c(y - \mu_y)^2)$$

and

$$f^c = \frac{\cos^2\theta^c}{2\sigma_x^c} + \frac{\sin^2\theta^c}{2\sigma_y^c},$$
$$g^c = \frac{\sin2\theta^c}{4\sigma_y^c} - \frac{\sin2\theta^c}{4\sigma_y^c},$$
$$h^c = \frac{\sin^2\theta^c}{2\sigma_x^c} + \frac{\cos^2\theta^c}{2\sigma_y^c},$$

and likewise for $G^s$. We initialised $(\mu_x, \mu_y)$ in the following way: Since we set the model readout's location parameters to (0, 0) for all model neurons when generating their MEIs, we also expected the MEIs to be centred at (0, 0), as well. Hence, we determined the location of the minimum and the maximum value of the MEI; whichever was closer to the centre (0,0) provided the initial values for the parameters $(\mu_x, \mu_y)$. Starting from there, we then first fit a single Gaussian to the MEI, and took the resulting parameters as initial parameters for the DoG fit. This was a constrained optimisation problem, with lower and upper bounds on all parameters; in particular, such that the location parameter would not exceed the canvas of the MEI, and such that the variance would be strictly positive.

## MEI properties

### Centre size

We defined the diameter of the centre of the MEI in the horizontal and the vertical orientation, respectively, as $d_x^c = 2\sigma_x^c$ and $d_y^c = 2\sigma_y^c$. The centre size was calculated as $\frac{1}{2}(d_x^c + d_y^c)$. We then estimated a contour outlining the MEI centre as the line that is defined by all points at which the 2D centre Gaussian $G^c$ attains the value $G^c(x, y)$ with $(x, y) = (\mu_x + \sigma_x^c, \mu_y + \sigma_y^c)$. The centre mask $m$ was then defined as a binary matrix with all pixels within the convex hull of this contour being 1 and all other pixels set to 0. This mask is used for calculating centre chromatic contrast (see below).

### Temporal frequency

To estimate temporal frequency of the MEIs, we estimated the power spectrum of the temporal components using a fast Fourier transform after attenuating high frequency noise by filtering with a fifth-order low-pass Butterworth filter with cutoff frequency 10 Hz. We then estimated the mean frequency of the temporal component by calculating an average of the frequency components, each weighted with its relative power.

### Contrast

The contrast of the MEIs in the two channels, $\gamma(\boldsymbol{x}_c^{*(n)})$ for $c \in [\text{green, UV}]$, was defined as the difference between the mean value within the centre mask $m$ at the two last peaks of the temporal component of the MEI in the UV channel at time points $t_2$ and $t_1$:

$$\gamma(\boldsymbol{x}_c^{*(n)}) = (\boldsymbol{x}_c^{*(n)} \odot m)(t_2) - (\boldsymbol{x}_c^{*(n)} \odot m)(t_1),$$

where $\odot$ denotes the element-wise multiplication of the MEI and the binary mask (see *Figure 3f*). The peaks were found with the function `scipy.signal.find_peaks`, and the peaks found for the UV channel were used to calculate contrast both in the green and the UV channel.

## Validating MEIs experimentally

### Generating MEI stimuli

To test experimentally whether the model correctly predicts which stimuli would maximally excite RGCs of different RGC groups, we performed a new set of experiments (numbers indicated in red in *Figure 1—figure supplement 1*), where we complemented our stimulus set with MEI stimuli. For the MEI stimuli, we selected 11 RGCs, chosen to span the responses space and to represent both well-described and poorly understood RGC groups, for which we generated MEIs at different positions on a 5 × 5 grid (spanning 110 µm in vertical and horizontal direction). We decomposed the MEIs as described above, and reconstructed MEIs as rank 1 tensors by taking the outer product of the spatial and temporal components:

$$\bar{x}^* = S_{11} U_{:1} \otimes V_{:1}^T$$

The MEI stimuli, lasting 50 frames (1.66 s) were padded with 10 frames (0.34 s) of inter-stimulus grey, and were randomly interleaved. With 11 stimuli, presented at 25 positions and lasting 2 s each, the total stimulus duration was 11 × 25 × 2 s = 550 s. Since the model operated on a z-scored (0 mean, 1 SD) version of the movie, MEIs as predicted by the model lived in the same space and had to be transformed back to the stimulator range ([0, 255]) before being used as stimuli in an experiment by scaling with the movie's SD and adding the movie's mean. The MEIs' green channel was then displayed with the green LED, and the UV channel was displayed with the UV LED. For experiments at Northwestern University, an additional transform was necessary to achieve the same levels of photoreceptor activation (photoisomerisation rates) for M- and S-cones with different LEDs. To ensure proper chromatic scaling between the different experimental apparatuses with different spectral profiles, we described the relative activation of M- and S-cones by the green and UV LEDs in the stimulation setup used in the two-photon imaging experiments (setup **A**) by a matrix

$$\mathbf{A} = \begin{bmatrix} a_{mg} & a_{sg} \\ a_{mu} & a_{su} \end{bmatrix} = \begin{bmatrix} 1 & 0.19 \\ 0 & 1 \end{bmatrix},$$

and the relative activation of M- and S-cones by the stimulation setup used in the patch-clamp experiments (setup **B**) by a matrix

$$\mathbf{B} = \begin{bmatrix} b_{mg} & b_{sg} \\ b_{mu} & b_{su} \end{bmatrix} = \begin{bmatrix} 1 & 0.9 \\ 0.035 & 1 \end{bmatrix},$$

where diagonal entries describe the activation of M-cones by the green LED, and of S-cones by the UV LED, and entries in the off-diagonal describe the cross-activation (i.e. M-cones by UV-LED and S-cones by green LED). The activation of M-cones and S-cones $e^T = (e_m, e_s)$ by a stimulus $x \in \mathcal{R}^{2 \times 1}$ displayed on a given stimulation setup was approximated as $\mathbf{e} = \mathbf{A}x$ (*Christenson et al., 2022*). Hence, a stimulus $x'$ displayed on setup **B**, defined as $x' = \mathbf{B}^{-1}\mathbf{A}x$, will achieve the same photoreceptor activation as stimulus $x$ displayed on setup **A**. Since the solution exceeded the valid range of the stimulator ([0, 255]), we added an offset and multiplied with a scalar factor to ensure all stimuli were within the valid range.

### Analysing RGC responses to MEI stimuli

We wanted to evaluate the responses of RGCs to the MEI stimuli in a spatially resolved fashion, i.e., weighting responses to MEIs displayed at different locations proportional to the strength of the RGCs RF at that location. In order to be able to meaningfully compare MEI responses between RGCs and across groups, for each RGC, we first centred and scaled the responses to zero mean and a standard deviation of 1. Then, for each RGC $n$, we computed a spatial average of its responses, weighting its responses at each spatial location $(x, y)$ proportional to the Gaussian density $\mathcal{N}_{\mu_n, \sigma_n}(x, y)$, where the parameters of the Gaussian $\mu_n = (\mu_x, \mu_y), \sigma_n$ were the model's estimated readout parameters for neuron $n$ (*Figure 4b, c, and d*, left):

$$\langle \boldsymbol{r}^{(n)} \rangle_{x,y} = \sum_{x'=1}^{5} \sum_{y'=1}^{5} \boldsymbol{r}^{(n)}_{x',y'} \cdot \mathcal{N}_{\boldsymbol{\mu_n}, \sigma_n}(x', y')$$

where $\boldsymbol{r}^{(n)}_{x',y'} \in \mathcal{R}^{11 \times 60}$ is the 60 frames (2 s) long response of neuron $n$ to an MEI at position $(x, y) = (x', y')$, resampled from the recording frame rate of 7.81 Hz to 30 Hz. We then averaged $\langle \boldsymbol{r}^{(i)} \rangle_{x,y}$ across time in the optimisation time window, i.e., frames 30–50, to get a scalar response $\tilde{r}^{(n)} = \langle \boldsymbol{r}^{(n)} \rangle_{x,y,t}$ for each MEI stimulus (***Figure 4d***).

## Selectivity index

To quantify the selectivity of the response $\tilde{r}^{(n)}(\boldsymbol{x}_i^*)$ of an RGC $n$ to an MEI $\boldsymbol{x}_i^*$, we defined a selectivity index as follows. First, we standardised the responses $\tilde{r}^{(n)}$ across all MEIs by subtracting the mean and dividing by the standard deviation. The selectivity index of RGC group $G_g$ to MEI $\boldsymbol{x}_i^*$ was then defined as

$$\mathrm{SI}_g(\boldsymbol{x}_i^*) = \langle \tilde{r}^{(n)}(\boldsymbol{x}_i^*) - \frac{1}{10} \sum_{j=1}^{11} \delta_{ij} \tilde{r}^{(n)}(\boldsymbol{x}_j^*) \rangle_n,$$

where $\delta_{ij}$ is the Kronecker delta. In words, the SI is the difference (in units of SD response) between the response to the MEI of interest ($\boldsymbol{x}_i^*$) and the mean response to all other (10) MEIs, $\frac{1}{10} \sum_{j=1}^{11} \delta_{ij} \tilde{r}^{(n)}(\boldsymbol{x}_j^*)$, averaged across all cells $n$ belonging to the group of interest $G_g$.

## Characterising nonlinear processing of chromatic contrast

We wanted to analyse the tuning of $G_{28}$/tSbC RGCs to chromatic contrast and to this end, we mapped the model response and its gradient across a range of chromatic contrasts (***Figure 6***). Specifically, the MEIs have $d = 2 \times 18 \times 16 \times 50 = 28,800$ pixels and dimensions, 14,400 for each colour channel. Now let $x^{*(n)} \in \mathcal{R}^{1 \times 28800}$ be the cell's MEI estimated using the CNN model, with the first $d=14,400$ dimensions defining the green pixels and the remaining dimensions defining the UV pixels. Then for each cell, we first consider a two-dimensional subspace spanned by two basis vectors $\mathbf{e}_1, \mathbf{e}_2$ where

$$\mathbf{e}_1 = -s_1 \begin{bmatrix} x_1^{*(n)} \\ x_2^{*(n)} \\ \vdots \\ x_d^{*(n)} \\ 0 \\ \vdots \\ 0 \end{bmatrix} \qquad \mathbf{e}_2 = s_2 \begin{bmatrix} 0 \\ \vdots \\ 0 \\ x_d^{*(n)} \\ x_{d+1}^{*(n)} \\ \vdots \\ x_{2d}^{*(n)} \end{bmatrix}$$

In words, the first basis vector $\mathbf{e}_1$ consists of the green component of the MEI in the green channel, multiplied by $-1$, and of 0 s in the UV channel, and the second basis vector consists of the UV component in the UV channel and of 0 s in the green channel for $\mathbf{e}_2$. $s_1$ and $s_2$ are two scaling factors chosen to equalise contrast (as measured by L2 norm) across colour channels while preserving the contrast of the stimulus as a whole. In the subspace spanned by these basis vectors, the point $[-1, 1]$ represents a contrast-scaled version of the original MEI. We then sampled 11 points along each dimension, equally spaced between $[-1, 1]$, which resulted in stimuli that are identical in terms of their spatial and temporal properties and only differ in their contrast. We then evaluated the model neuron response at these points in the subspace (***Figure 6d***). We also evaluated the gradient of the model neuron response at these points and plotted the direction of the gradient projected into the subspace spanned by $e_1, e_2$ (***Figure 6b and c***).

## Detection performance analysis

To test the performance of individual RGCs of different groups in detecting the target class of inter-clip transitions (ground-to-sky) from all other classes of inter-clip transitions, we performed an ROC analysis (***Fawcett, 2006***). For each RGC, we calculated its response to an inter-clip transition

occurring at time $t_0$ as the baseline-subtracted average response within 1 s following the transition, i.e., $\frac{1}{T}\sum_{t=0}^{T} r(t) - r(t_0)$, with $T$=30 frames at 30 Hz. For all $N$=40 equally spaced thresholds within the response range of an RGC, we then calculated the TPR and FPR of a hypothetical classifier classifying all transitions eliciting an above-threshold response as a positive, and all other transitions as negative. Plotting the TPR as a function of FPR yields an ROC curve, the area under which (AUC) is equivalent to the probability that the RGC will respond more strongly to a randomly chosen inter-clip transition of the target class than to a randomly chosen inter-clip transition of a different class. The AUC thus is a measure of performance for RGCs in this detection task.

## Detection task in simulation

We simulated the four types of transitions (sky-sky, sky-ground, ground-ground, ground-sky) in natural scenes at different velocities, which could be triggered by different behaviours such as locomotion or eye movements. With the simulated context-changing stimuli, we predicted model neuron responses in silico and then determined if $G_{28}$ could perform the detection task robustly well across speeds.

For generating the stimuli, 500 frames were randomly extracted from the same mouse natural movies used for the 2P imaging experiments. For each frame, we simulated visual transitions by moving a 72×64 pixel-large window along a fixed trajectory (*Figure 7h*, bottom) at four different angular velocities: 50, 150, 250, and 350/s, corresponding to 4, 12, 20, and 28 pixels per frame, respectively (*Figure 7—figure supplement 3a and b*). Each edge of the trajectory is 220 pixels long, covering 90.6 of visual angle. Each selected scene frame was sampled eight times (i.e. twice per velocity). To avoid potential biases due to asymmetries in the mouse natural movie, we sampled each frame for each velocity both in clockwise and in counterclockwise direction. The stimuli were then down-sampled to 18×16 pixels and shown to the model at a frame rate of 30 Hz. Because the trajectories contained different numbers of moving frames for the four velocities, we 'padded' the stimuli at the beginning and the end of each transition stimulus by duplicating the start and end frames, resulting in a total of 60 frames each (see illustration in *Figure 7—figure supplement 3*).

## Statistical analysis

### Permutation test

We wanted to test how likely the difference in AUC observed for different RGC groups are to occur under the null hypothesis that the underlying distributions they are sampled from are equal. To this end, we performed a permutation test. We generated a null distribution for our test statistic, the absolute difference in AUC values $\Delta$AUC, by shuffling the RGC group labels of the two groups of interest (e.g. $G_{28}$ and $G_{24}$) and calculating the test statistic with shuffled labels 100,000 times. We only included RGC groups with at least $N$=4 cells in this analysis. We then obtained a p-value for $\Delta$AUC observed with true labels as the proportion of entries in the null distribution larger than $\Delta$AUC.

### Bootstrapped confidence intervals

We bootstrapped confidence intervals for $\Delta$AUC (*Figure 7* and *Figure 7—figure supplement 2*). For $\Delta$AUC, we generated a bootstrapped distribution by sampling 100 times with replacement from the AUC values of the two groups that were being compared and calculated $\Delta$AUC. We then estimated the 95% confidence interval for $\Delta$AUC as the interval defined by the 2.5th and 97.5th percentile of the bootstrapped distribution of $\Delta$AUC.

For $\Gamma(\phi_s, \phi_{\nu_g})$, we generated a bootstrapped distribution by sampling 100 times with replacement from the MEI responses of RGC group $g$ and then calculating $\text{RDM}^{\phi_{\nu_g}}$ and $\Gamma(\phi_s, \phi_{\nu_g})$ for each sample. We then estimated the 95% confidence interval for $\Gamma(\phi_s, \phi_{\nu_g})$ as the interval defined by the 2.5th and 97.5th percentile of the bootstrapped distribution of $\Gamma(\phi_s, \phi_{\nu_g})$.

### Estimating effect size

The effect size of difference in AUC observed for different RGC groups $l$ and $k$, $\Delta$AUC (*Figure 7* and *Figure 7—figure supplement 2*), was estimated as Cohen's $d$ (*Cohen, 1988*; *Goulet-Pelletier and Cousineau, 2018*):

$$\frac{|m_k - m_l|}{s},$$

with

$$s = \sqrt{\frac{(N_k - 1)s_k^2 + (N_l - 1)s_l^2}{N_k + N_l - 2}}$$

and $m_k$ and $s_k$ the sample mean and standard deviation, respectively, of the AUC observed for the $N_k$ RGCs of group k.

## Estimating linear correlation

Wherever the linear correlation between two paired samples $x$ and $y$ of size $N$ was calculated for evaluating model performance, *Figure 2*, *Figure 1—figure supplement 1*, *Figure 4—figure supplement 1*, we used Pearson's correlation coefficient:

$$C_{xy} = \frac{\sum_i^N (x_i - \bar{x})(y_i - \bar{y})}{\sqrt{\sum_i^N (x_i - \bar{x})^2}\sqrt{\sum_i^N (y_i - \bar{y})^2}}$$

## Acknowledgements

We thank Jonathan Oesterle and Dominic Gonschorek for feedback on the manuscript, Jan Lause for statistical consulting, and Merle Harrer for general assistance. We also thank all members of the Sinz lab for regular discussions on the project. This work was supported by the German Research Foundation (DFG; CRC 1233 'Robust Vision: Inference Principles and Neural Mechanisms', project number 276693517 to PB, MB, TE, KF; Heisenberg Professorship, BE5601/8-1 to PB; Excellence Cluster EXC 2064, project number 390727645 to PB; CRC 1456 'Mathematics of Experiment' project number 432680300 to ASE), the Federal Ministry of Education and Research (FKZ 01IS18039A to PB), National Institutes of Health (NIH; NEI EY031029, NEI EY031329 to GWS; NEI F30EY031565, NIGMS T32GM008152 to ZJ), and the European Research Council (ERC) under the European Union's Horizon Europe research and innovation programme (ASE, grant agreement No. 101041669). We acknowledge support from the Open Access Publication Fund of the University of Tübingen.

## Additional information

### Funding

| Funder | Grant reference number | Author |
| --- | --- | --- |
| Deutsche Forschungsgemeinschaft | CRC 1233 project number 276693517 | Matthias Bethge Philipp Berens Katrin Franke Thomas Euler |
| Deutsche Forschungsgemeinschaft | Heisenberg Professorship BE5601/8-1 | Philipp Berens |
| Deutsche Forschungsgemeinschaft | EXC 2064 390727645 | Philipp Berens |
| Deutsche Forschungsgemeinschaft | CRC 1456 project number 432680300 | Alexander S Ecker |
| Bundesministerium für Bildung und Forschung | FKZ 01IS18039A | Philipp Berens |
| National Institutes of Health | NEI EY031029 | Gregory W Schwartz |
| National Institutes of Health | NEI F30EY031565 | Zachary Jessen |
| European Research Council | grant agreement No. 101041669 | Alexander S Ecker |

| Funder | Grant reference number | Author |
|---|---|---|
| National Institutes of Health | NEI EY031329 | Gregory W Schwartz |
| National Institutes of Health | NIGMS T32GM008152 | Zachary Jessen |

The funders had no role in study design, data collection and interpretation, or the decision to submit the work for publication.

## Author contributions

Larissa Höfling, Conceptualization, Data curation, Software, Formal analysis, Validation, Visualization, Methodology, Writing - original draft, Writing – review and editing; Klaudia P Szatko, Data curation, Validation, Investigation; Christian Behrens, Software, Formal analysis, Methodology; Yuyao Deng, Validation, Visualization, Methodology, Writing – review and editing; Yongrong Qiu, Resources, Software; David Alexander Klindt, Software, Methodology, Writing – review and editing; Zachary Jessen, Validation, Investigation, Visualization, Methodology, Writing - original draft; Gregory W Schwartz, Supervision, Funding acquisition, Validation, Investigation, Methodology, Writing - original draft; Matthias Bethge, Philipp Berens, Katrin Franke, Alexander S Ecker, Conceptualization, Supervision, Funding acquisition, Writing – review and editing; Thomas Euler, Conceptualization, Supervision, Funding acquisition, Visualization, Writing - original draft, Project administration, Writing – review and editing

## Author ORCIDs

Larissa Höfling (iD) http://orcid.org/0000-0003-2459-0706
Christian Behrens (iD) https://orcid.org/0000-0003-3623-352X
Gregory W Schwartz (iD) https://orcid.org/0000-0001-8909-4397
Matthias Bethge (iD) https://orcid.org/0000-0002-6417-7812
Philipp Berens (iD) https://orcid.org/0000-0002-0199-4727
Katrin Franke (iD) https://orcid.org/0000-0002-8649-4835
Alexander S Ecker (iD) https://orcid.org/0000-0003-2392-5105
Thomas Euler (iD) https://orcid.org/0000-0002-4567-6966

## Ethics

All imaging experiments were conducted at the University of Tübingen; the corresponding animal procedures were approved by the governmental review board (Regierungspräsidium Tübingen, Baden-Württemberg, Konrad-Adenauer-Str. 20, 72072 Tübingen, Germany) and performed according to the laws governing animal experimentation issued by the German Government. All electrophysiological experiments were conducted at Northwestern University; the corresponding animal procedures were performed according to standards provided by Northwestern University Center for Comparative Medicine and approved by the Institutional Animal Care and Use Committee (IACUC).

## Decision letter and Author response

Decision letter https://doi.org/10.7554/eLife.86860.sa1
Author response https://doi.org/10.7554/eLife.86860.sa2

# Additional files

## Supplementary files
• MDAR checklist

## Data availability

The data and the movie stimulus are available at https://gin.g-node.org/eulerlab/rgc-natstim. Code for training models and reproducing analyses and figures is available at https://github.com/eulerlab/rgc-natstim-model, (copy archived at *Hoefling et al., 2024*).

The following dataset was generated:

| Author(s) | Year | Dataset title | Dataset URL | Database and Identifier |
|---|---|---|---|---|
| Höfling H, Szatko KP, Behrens C, Yeng D, Qiu Y, Klindt DA, Jessen Z, Schwartz GM, Bethge M, Berens P, Franke K, Ecker AS, Euler T | 2024 | rgc-natstim-model | https://gin.g-node.org/eulerlab/rgc-natstim | g-node, rgc-natstim-model |

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
