## [Editor Report]

This study presents a fundamental and very technically strong dataset of mouse ganglion cells responding to natural stimuli that include more natural chromatic properties. Fits of convolutional neural networks to experimental measurements highlighted a novel form of color opponency in suppressed-by-contrast ganglion cells. More generally, the work provides a compelling example of how modern experimental and computational tools can be used to generate and test hypotheses about sensory function under natural conditions.

---

## [Decision Letter]

**Decision letter after peer review:**

Thank you for submitting your article "A chromatic feature detector in the retina signals visual context changes" for consideration by *eLife*. Your article has been reviewed by 3 peer reviewers, one of whom is a member of our Board of Reviewing Editors, and the evaluation has been overseen by Lois Smith as the Senior Editor.

All reviewers were impressed by the technical aspects of the work. Three key issues emerged in review and were emphasized in discussions among the reviewers. First, the MEI consists of a single linear visual feature to represent a neuron's spatial and temporal stimulus selectivity. Such a linear feature cannot capture many key nonlinear aspects of retinal signaling (e.g. direction selectivity among many others). This limitation needs to be clearer in the paper. It is, for example, important that a reader does not come away thinking that the paper provides a general approach to identifying stimulus selectivity. Second, the MEIs in some ways resemble more standard measures of receptive field properties and in others differ strongly. A more complete comparison is needed of MEIs vs. standard measures, as well as a discussion of why they might differ (which could be related to the first main point above). Third, the paper emphasizes the possible role of these cells in horizon detection, but the analyses that support this role are based on ground-to-sky transitions, and the horizon itself was excluded from the stimulus set. Hence it does not seem possible to evaluate the role of horizon detection. It is important that discussions of ethological function are more grounded in the analyses provided in the paper.

*Reviewer #1 (Recommendations for the authors):*

Line 170-171: Suggest modifying the sentence "This established that …" to be more objective. It's not clear that a correlation of 0.5 supports that sentence.

Line 173: Isn't the linearized CNN equivalent to a straight linear filter model? If so I would state that – it's a simpler description. If not, I would clarify how it is different.

Line 357-359: What is the indication that the warping is in the color opponent region?

*Reviewer #2 (Recommendations for the authors):*

1. Line 856. Why were clips restricted to a certain mean intensity range, and how did this influence the analysis? The concern is that allowing a more natural range of means would then violate the sky-ground transition specificity, as cells might fire in a broader range of conditions.

2. Figure S2. Add UV / Green labels to the two grayscale maps as in Figure 4a.

3. Figure 2A. Number of channels should be added to the figure.

4. Figure 2A. 'Receptive fields' label is only indicated in the last layer but is defined by the whole network. This is confusing, perhaps for the last layer something should be labelled with something more specifically relating to spatial localization?

5. Figure 2A. Is the Kronecker (or 'tensor') product symbol really intended? If so, please state so in the legend and clarify in methods how this relates to convolution implementation. If it is just generally meant to be convolution, a different symbol should be used and stated. In other words, it helps understanding to be formally correct with symbols, with more formalized mathematics or physics usage preferred to less regulated machine learning usage.

6. Figure 2B. How can so many cells be above the maximum set by the cell's response reliability?

7. Figure 2c. A 'linear model' is stated, but from the methods, it seems like an LN model (the final threshold is still present). If it really is a linear model, this is an inappropriate straw man (a threshold should be included) but if it is an LN model, that should be stated correctly.

8. Line 240 -241. Due to "the fixed contrast budget across channels", the contrast of most MEIs is shifted toward the UV. Please explain what is meant by the fixed contrast budget and its effects.

9. Figure 4G. The block structure of Figure 4G is hard to interpret, because it is not clear what the expectation is. For many of the cells, the MEI is not the most effective input, not even for the model. Attention is focused on the one cell that has the most distinctive response to its MEI, but is this just a random occurrence? Some guidance clarifying the interpretation of this confusion matrix would be helpful, or whether it is just interesting but we are not supposed to have any real interpretation of it.

10. Line 255. If only 2/3 of the cells have color opponent MEIs, are they really a cell class? Does this call into question the classification approach or the MEI approach?

11. Related to the difficulties of the representation analysis, it is not clear what 'sacrifice' occurs in favor of chromatic discrimination. If cell 28 did not exist, some chromatic discrimination would be lost, but other cell types would still be there to represent other features.

12. Figure 7d. the retinal cross-section image is confusing. Due to retinal warping, the ChAT bands droop below the IPL boundaries indicated, and therefore the arborization level seen in the maximal projection is misleading. It would be better to use the ChAT bands to correct for warping at each spatial location so that the dendritic arborization could be interpreted.

13. Line 609. This section relates to a previous claim, summarized as 'the dumber the animal the smarter the retina'. This is regarded as a myth, and experimental evidence does not support this (see Gollisch and Meister, 2010 discussion, point 2). It would be better for the field not to revive this idea.

14. Line 624. There is a misstatement about how this paper contributes to 'how' a computation is implemented, as there is no mechanistic information here about the circuitry.

15. Line 871. Please make clear that the two channels are chromatic, and a different word should be used than 'channel' because the layer has 16 channels by the usual meaning.

16. Line 874. More detail should be given about the Fourier parameterization.

17. The limitations of the slow frame rate should be more clearly acknowledged.

[Editors' note: further revisions were suggested prior to acceptance, as described below.]

Thank you for resubmitting your work entitled "A chromatic feature detector in the retina signals visual context changes" for further consideration by *eLife*. Your revised article has been evaluated by Lois Smith (Senior Editor), a Reviewing Editor and the three original reviewers.

All the reviewers agreed that the manuscript has improved in revision and appreciated the new data and analyses that were added. There are a few remaining issues that need to be addressed, as outlined in the reviews below. Most important is clarifying the limitations of the MEI analysis earlier in the paper. These are detailed in the individual reviews below.

*Reviewer #1 (Recommendations for the authors):*

This paper has improved with the new analyses and revisions added in response to reviews. These changes have clarified several issues, provided stronger evidence for others, and brought the results and text closer in alignment. A few issues have either emerged or been highlighted in the revision:

MEI and temporal tuning. The text at times could be read as saying that the MEIs reflect a better way to measure tuning properties. A concern about this terminology is the dependence of the MEI on the approach used to optimize responses (e.g. on the time window used in that approach). For example, the temporal frequencies that contribute to the time course of the MEI are quite low – likely reflecting the optimization procedure. One suggestion is to clarify in the section starting on line 204 that the MEIs reflect a combination of a cell's own properties and the optimization procedure (as you have already for the oscillations in MEIs for some cells).

Figure 6b, c: What do the contrasts beyond -1 or 1 mean?

Figure 6e, f: This cell does not look opponent, with both UV and green producing On responses. Is this the only cell this experiment was performed on? If not, were other cells more in alignment with the predictions from the model?

Figure 7i: The colors are hard to see and match to those in the rest of the figure. In addition, the gray bars and labels should be defined in the caption.

*Reviewer #2 (Recommendations for the authors):*

The paper has improved but still does not acknowledge its limitations sufficiently.

1. With respect to the difference between the MEI and the linear receptive field, there was no misunderstanding in the previous review, and the current manuscript still does not acknowledge the limitations of the MEI analysis. This point could have been made more explicit in the previous review, and so the following is an attempt to do so.

By analogy, in a topographic map of the world, the MEI is the location of Mt. Everest. The linear receptive field is a plane fit to the direction of the steepest ascent fit to the world map, additionally including an average slope. A Linear-Nonlinear model allows variation in height along that single direction. But a full understanding between the relationship between change in spatial position and height requires the full two-dimensional map.

In the retina, and in a CNN model of the retina, parallel rectified pathways create sensitivity to many more different directions in stimulus space, commonly referred to as features. Each separate interneuron with a threshold potentially creates a different feature or dimension, with firing rate and sensitivity varying in each point of the high-dimensional space spanned by the set of features.

Linear-Nonlinear models and MEIs only encode sensitivity to a single feature. However, multiple distinct features are required to produce a wide set of important phenomenon, including nonlinear subunits that cause sensitivity to fine textures, direction selectivity for both light and dark objects (On-Off), object motion sensitivity, latency coding, the omitted stimulus response, responses to motion reversal, and pattern adaptation (reviewed in Gollisch and Meister, 2010). All of these properties rely on multiple distinct features and their interactions. Any analysis based solely on an MEI necessarily abandons consideration of how sensitivity along these different features combines to produce a computation. The new Figure 6c, which examines two stimulus directions near the MEI is an improvement, but it is only two directions in one region of stimulus space.

There is substantial concern that readers will misinterpret the 'Most Exciting' input to mean the 'Most Important'. It would therefore serve the field for a clear statement to be made that analysis of MEIs alone (1) will not capture interactions of multiple nonlinear neural pathways, (2) therefore will not capture nonlinear interactions between multiple stimulus features, and (3) will consequently not explain the set of phenomena that rely on these neural pathways and stimulus features.

In response to the authors rebuttal, to clarify the nonlinear analysis that one might do beyond the analysis of MEIs to identify multiple dimensions, these would include Spike-Triggered Covariance (Fairhall et al., 2006), Maximally Informative Dimensions (Sharpee et al., 2003), Maximum Noise Entropy (Globerson et al., 2009), Nonnegative Matrix Factorization (Liu et al., 2017), proximal algorithms (Maheswaranathan et al., 2018), and model reduction approaches to reduce to the CNN model to fewer dimensions (Maheswaranathan et al., 2023) among others.

2. It should be made explicit for each analysis whether it is based on measured responses or stimuli presented to the model. In Figure 7, a proper signal detection analysis cannot be performed without accounting for noise on single trials. It is unclear whether analyzed responses are directly from data, or from stimuli presented to the model. For extrapolation to different speeds, these are clearly from new stimuli presented to the model. It should be stated that without accounting for measured noise, ROC analyses will overestimate detection of ground-sky transitions.

---

## [Author Response]

Essential revisions:All reviewers were impressed by the technical aspects of the work. Three key issues emerged in review and were emphasized in discussions among the reviewers.

We thank editors and reviewers for their constructive input.

First, the MEI consists of a single linear visual feature to represent a neuron's spatial and temporal stimulus selectivity. Such a linear feature cannot capture many key nonlinear aspects of retinal signaling (e.g. direction selectivity among many others). This limitation needs to be clearer in the paper. It is, for example, important that a reader does not come away thinking that the paper provides a general approach to identifying stimulus selectivity.Second, the MEIs in some ways resemble more standard measures of receptive field properties and in others differ strongly. A more complete comparison is needed of MEIs vs. standard measures, as well as a discussion of why they might differ (which could be related to the first main point above).

We think that there is a misunderstanding about the differences between the MEI approach and the more standard measures of receptive field (RF) properties the reviewers refer to. We apologize if this was not clear enough in the original manuscript.

In the following, we will first explain the differences between MEIs and linear RF estimates in more detail. We then show a new analysis (also included in the revised manuscript) that shows more clearly that for G28/tSbC cells (the chromatic feature detectors described in our paper) the MEI is not the same as the linear filter. In fact, the linear filter does not recover the chromatic selectivity reliably, and hence a nonlinear approach such as the MEIs is necessary to characterize G28/tSbC RGCs adequately.

The fundamental difference between more standard measures of receptive field (RF) properties and MEIs is the following: Classical approaches for estimating the RF typically aim at capturing a cell’s linear filter. The estimated linear filter is an approximation of how the cell processes incoming stimuli; RF estimates can therefore be used as linear filters, for example in Linear-Nonlinear (LN) models that predict cell responses to stimuli. The MEI approach, conversely, is an approach for estimating the optimal stimulus for a cell: an optimisation procedure is leveraged to maximize an objective function, which, in the simplest case, can be the model neuron response prediction. However, the objective function can be any function of the predicted response. Importantly, we would like to point out that MEIs do not simply approximate nonlinear stimulus selectivity by a linear filter, but instead are the image that leads to the peak in the nonlinear activation landscape, which is why they are well-suited for characterizing nonlinear systems, such as the retina. For a truly linear system, the two coincide, but not for the retina, because, as the reviewers rightly point out, the retina is nonlinear. Crucially, the MEI approach can also recover stimulus selectivity if a cell is selective for a combination of stimulus features, since the combination of features can be described as a point in high-dimensional stimulus space.

To demonstrate the necessity of a non-linear approach (i.e., MEIs) for recovering the chromatic opponency of RGC type G28, we tested whether we could recover this feature using an LN model (i.e., a CNN model with no nonlinearities between convolutional layers, but with a final thresholding nonlinearity in place). Indeed, with the LN model, only 9 out of 36 G28 cells were identified as color-opponent (nonlinear model: 24 out of 36; see new Figure 6a). This result demonstrates that the preference for chromatic contrast of G28 cells does not result from a color opponent linear filter (i.e. negative weights for green, and positive weights for UV), which could also be captured by an LN-type model. Rather, there must be a nonlinear dependence between chromatic contrast (of the stimulus) and chromatic selectivity (of the cell).

To understand the nature of this dependence, we further expanded the analysis by estimating the model neurons’ tuning to color contrast around the maximum (the MEI). To this end, we mapped the model neurons’ response in 2D chromatic contrast space (new Figure 6b-d), revealing that, indeed, G28/tSbC RGCs have a nonlinear tuning for color contrast: their selectivity for color contrast changes depending on the location in the space of color contrast. We verified these model predictions by electrically recording from morphologically identified G28/tSbC cells (see new Figure 6e,f, also above).

In summary, in the revised manuscript, we (a) added a more detailed explanation of the difference between more conventional linear approaches for estimating stimulus selectivity on the one hand, and the MEI approach as a more general approach to identify nonlinear stimulus selectivity on the other hand. We also offered intuitions on how to interpret the MEIs, especially for those features that differ from linear RF measures (e.g. the temporal oscillations). We also (b) discussed the limitations of the MEI approach (see also replies to individual reviewer comments below). Further, we (c) added an analysis (and new electrophysiological data) of the chromatic contrast selectivity in tSbC RGC, demonstrating that this is a nonlinear feature, which could only be recovered using a nonlinear model.

Third, the paper emphasizes the possible role of these cells in horizon detection, but the analyses that support this role are based on ground-to-sky transitions, and the horizon itself was excluded from the stimulus set. Hence it does not seem possible to evaluate the role of horizon detection. It is important that discussions of ethological function are more grounded in the analyses provided in the paper.

We acknowledge that we were not clear enough about the proposed function for this cell type. What we suggest is that G28/tSbC cells contribute to detecting context changes which are marked by a change in color contrast, such as transitions across the horizon and specifically, from ground to sky. We do not think that the cells are detecting the horizon itself. We improved the revised manuscript in this respect.

To back-up our hypothesis that G28/tSbC cells play a role in detecting context changes, we performed an additional simulation analysis. We sampled 1,000 stimuli from the original mouse movie (Qiu et al. Curr Biol 2021) by moving a window across a scene (Figure 7h), mimicking the kind of stimulus elicited by eye, head, and/or whole body movement. As before, each stimulus mimicked one of four transition types: ground->ground, sky->sky, sky->ground, and ground->sky. The former two transitions correspond to changes in visual input without context change, whereas the latter two reflect a context change. Importantly, the ground->sky and sky->ground transitions in these simulations contain the horizon.

We then simulated responses for all 32 RGC groups and performed an ROC analysis (analogous to the one already described in the manuscript) to determine how well the different RGC groups could distinguish ground->sky transitions from the other transitions.

To test robustness of detection performance across different speeds of transitions, we performed this analysis at different angular velocities (50, 150, 250, and 350°/s; see Figure 6-supplement 3a,b). Interestingly, while most slow ON RGC types improve their performance with increasing speeds, G28 performs well robustly across all four speeds.

In the revised manuscript, we added this new analysis (cf. Figure 7h-j, Figure 6-supplement 3) and discuss the implications of these results in the Discussion.

Reviewer #1 (Recommendations for the authors):Line 170-171: Suggest modifying the sentence "This established that …" to be more objective. It's not clear that a correlation of 0.5 supports that sentence.

We removed the sentence.

Line 173: Isn't the linearized CNN equivalent to a straight linear filter model? If so I would state that – it's a simpler description. If not, I would clarify how it is different.

The linearised CNN is equivalent to an LN model. We now state this in the manuscript and refer to this model as the “LN model”.

Line 357-359: What is the indication that the warping is in the color opponent region?

We removed the referenced section from the manuscript (see also above). However, it is visible from Figure 6b-d, and Figure 5-supplement 1 that the warping is in the color opponent region.

Reviewer #2 (Recommendations for the authors):1. Line 856. Why were clips restricted to a certain mean intensity range, and how did this influence the analysis? The concern is that allowing a more natural range of means would then violate the sky-ground transition specificity, as cells might fire in a broader range of conditions.

In a pilot study with unrestricted mean intensity range, we found that strong differences in mean intensity between clips dominated RGC responses, so we restricted the mean intensity range. For our additional analysis (Figure 7h-j), however, we did not restrict the range of intensities, and still found G28/tSbC cells to be selective for ground-sky transitions.

2. Figure S2. Add UV / Green labels to the two grayscale maps as in Figure 4a.

Done.

3. Figure 2A. Number of channels should be added to the figure.

Done.

4. Figure 2A. 'Receptive fields' label is only indicated in the last layer but is defined by the whole network. This is confusing, perhaps for the last layer something should be labelled with something more specifically relating to spatial localization?

We changed the label to “spatial RF”.

5. Figure 2A. Is the Kronecker (or 'tensor') product symbol really intended? If so, please state so in the legend and clarify in methods how this relates to convolution implementation. If it is just generally meant to be convolution, a different symbol should be used and stated. In other words, it helps understanding to be formally correct with symbols, with more formalized mathematics or physics usage preferred to less regulated machine learning usage.

Thanks for pointing that out; we replaced the symbol by that for a convolution.

6. Figure 2B. How can so many cells be above the maximum set by the cell's response reliability?

This is because the test set correlation and the response reliability are calculated in different ways (see Methods) such that their absolute values are not directly comparable. We expect a correlation between the two metrics, which the panel is meant to illustrate, but not a direct correspondence in terms of absolute values.

7. Figure 2c. A 'linear model' is stated, but from the methods, it seems like an LN model (the final threshold is still present). If it really is a linear model, this is an inappropriate straw man (a threshold should be included) but if it is an LN model, that should be stated correctly.

We apologize for the confusion, as stated above, it is indeed a CNN model equivalent to an LN model. We now clearly state that in the paper and changed the label to “LN model”.

8. Line 240 -241. Due to "the fixed contrast budget across channels", the contrast of most MEIs is shifted toward the UV. Please explain what is meant by the fixed contrast budget and its effects.

This statement was slightly misleading; the underlying reason for the UV-shift in MEI contrast is the dominance of UV-sensitive S-opsin in the ventral retina, making stimuli in the UV channel much more effective at eliciting RGC responses. The sharing of the fixed contrast budget across channels (see also Franke et al. 2022, Nature) was the *methodological* feature that led to this biological fact being reflected in the MEIs. In other words, the MEIs generated like this are the optimal stimuli, *conditioned on having a shared fixed contrast budget*. The optimal stimuli conditioned on having separate contrast budgets for the two color channels are expected to look different, i.e. with more contrast in the green channel.

We updated the Results section accordingly.

9. Figure 4G. The block structure of Figure 4G is hard to interpret, because it is not clear what the expectation is. For many of the cells, the MEI is not the most effective input, not even for the model. Attention is focused on the one cell that has the most distinctive response to its MEI, but is this just a random occurrence? Some guidance clarifying the interpretation of this confusion matrix would be helpful, or whether it is just interesting but we are not supposed to have any real interpretation of it.

We have a paragraph in the manuscript (original as well as revised) dedicated to this panel. In brief, we wanted to illustrate that RGCs show a strong response to their own group’s MEI, a weaker response to the MEIs of functionally related groups, and no response to MEIs of groups with different response profiles (Figure 4g top). Conversely, RGC groups from opposing regions in response space showed no response to each others’ MEIs. The model’s predictions showed a similar pattern (Figure 4g, bottom), thereby validating the model’s ability to generalize to the MEI stimulus regime.

10. Line 255. If only 2/3 of the cells have color opponent MEIs, are they really a cell class? Does this call into question the classification approach or the MEI approach?

We think that this is due to the classifier we employed, which did not use chromatic selectivity for classification (but chirp and moving bar responses, as well as soma size and direction/orientation selectivity). It was trained on the original dataset (Baden et al.. Nature 2016), where we decided to merge two clusters (38 and 39) to G28 (see Figure 2a there). This was done because the clusters only differed slightly in their direction and orientation selectivity index and we wanted to define functional RGC groups conservatively. Therefore, it is possible that G28 may collect other cells with similar chirp and moving bar responses but no color opponency. Similarly, the fact that G27 – with very similar chirp and moving bar responses – contains some color-opponent cells may also be explained by the original classifier.

11. Related to the difficulties of the representation analysis, it is not clear what 'sacrifice' occurs in favor of chromatic discrimination. If cell 28 did not exist, some chromatic discrimination would be lost, but other cell types would still be there to represent other features.

Our argument here is in terms of resources spent on a cell that monitors a particular region of stimulus space, while not monitoring the rest of stimulus space. In other words, we agree with the reviewer: If G28/tSbC cells did not exist, chromatic discrimination would be lost – but its existence suggests that the resources spent on it are relevant. However, as discussed above, we removed the referenced section from the manuscript.

12. Figure 7d. the retinal cross-section image is confusing. Due to retinal warping, the ChAT bands droop below the IPL boundaries indicated, and therefore the arborization level seen in the maximal projection is misleading. It would be better to use the ChAT bands to correct for warping at each spatial location so that the dendritic arborization could be interpreted.

We replaced the image by one of a different tSbC cell, where the tissue displayed less warping.

13. Line 609. This section relates to a previous claim, summarized as 'the dumber the animal the smarter the retina'. This is regarded as a myth, and experimental evidence does not support this (see Gollisch and Meister, 2010 discussion, point 2). It would be better for the field not to revive this idea.

There may not be experimental data supporting the idea that a larger “cortical” computational capacity results in “simpler” retinal receptive fields, but we think that recent modeling results (Lindsay et al. ICLR 2019) are compelling enough to consider this idea. Nonetheless, we amended this part of the discussion.

14. Line 624. There is a misstatement about how this paper contributes to 'how' a computation is implemented, as there is no mechanistic information here about the circuitry.

Changed.

15. Line 871. Please make clear that the two channels are chromatic, and a different word should be used than 'channel' because the layer has 16 channels by the usual meaning.

Thanks for pointing this out – we changed it to “input channels (green and UV)”, since the two color channels are the input channels in our case.

16. Line 874. More detail should be given about the Fourier parameterization.

Thanks for pointing this out – we changed it to “input channels (green and UV)”, since the two color channels are the input channels in our case.

17. The limitations of the slow frame rate should be more clearly acknowledged.

We are not sure to which specific limitations the reviewer refers. In any case, despite the low frame rate, the Ca^2+^ responses can be related to the spike rate (e.g., Román Rosón et al. Neuron 2019; Trapani et al. J Neurophysiol 2023, see their Figure Suppl. 4). We added a sentence to the Methods section.

[Editors’ note: what follows is the authors’ response to the second round of review.]

All the reviewers agreed that the manuscript has improved in revision and appreciated the new data and analyses that were added. There are a few remaining issues that need to be addressed, as outlined in the reviews below. Most important is clarifying the limitations of the MEI analysis earlier in the paper. These are detailed in the individual reviews below.Reviewer #1 (Recommendations for the authors):This paper has improved with the new analyses and revisions added in response to reviews. These changes have clarified several issues, provided stronger evidence for others, and brought the results and text closer in alignment. A few issues have either emerged or been highlighted in the revision:

We thank the reviewer for appreciating our revisions

MEI and temporal tuning. The text at times could be read as saying that the MEIs reflect a better way to measure tuning properties. A concern about this terminology is the dependence of the MEI on the approach used to optimize responses (e.g. on the time window used in that approach). For example, the temporal frequencies that contribute to the time course of the MEI are quite low – likely reflecting the optimization procedure. One suggestion is to clarify in the section starting on line 204 that the MEIs reflect a combination of a cell's own properties and the optimization procedure (as you have already for the oscillations in MEIs for some cells).

Thank you for pointing out that we came across as implying that MEIs are somehow “better” than other approaches. That is not what we wanted to imply. It is one method of many that can reveal nonlinear tuning properties and we used it to reveal a novel such tuning property. In the revised version of the manuscript, we have carefully rephrased all places where this impression might have been created.

Figure 6b, c: What do the contrasts beyond -1 or 1 mean?

For this analysis, we created visual stimuli in the two-dimensional subspace spanned by the MEI’s green and UV channel, respectively. In other words, we used the MEI’s green and UV channel as basis vectors and took a linear combination with different weights. These weights represent green and UV contrast, respectively (with respect to the MEI’s contrast). We then computed the predicted response of the model to these stimuli, which is shown in the figure. Note that we flipped the sign of the green channel’s basis vector, such that a positive weight corresponds to an ON-type stimulus and a negative weight to an OFF-type stimulus. That means the MEI is at (–1,1) in this plot.

In the previous visualisation, the axes were labelled to indicate the absolute contrast of the resulting stimuli, which could be outside the range (-1, 1), depending on the original MEIs contrast. We now realise that contrasts outside the -1…1 range may be confusing. We have therefore changed the axis labels to indicate the multiplicative scalar, and hence range from -1 to 1. We clarify this in the legend and explain in more detail in the methods section. In addition, we now use the same colour schemes for panels (b, c) and (e).

Note that, in the previous version of the manuscript, we chose a slightly different subspace (spanned by the UV component of the MEI both in the green and UV channel). For ease of interpretability, we now changed to the subspace described above. As a consequence, the panels in Figure 6 look slightly different.

Figure 6e, f: This cell does not look opponent, with both UV and green producing On responses. Is this the only cell this experiment was performed on? If not, were other cells more in alignment with the predictions from the model?

We thank the reviewer for this question and the opportunity to clarify. The colour opponency of this cell derives from a suppressive effect of green contrast, which can be best understood when comparing responses to a UV-ON, green-OFF stimulus (top left in panel (e); upper row, second from left in (f)) with the responses to a UV-ON, green-ON stimulus (top right in Figure 6 panel (e); upper row, far right in Figure 6. panel (f)).

In the manuscript, we previously wrote:

“This analysis revealed that, indeed, G28 RGCs have a nonlinear tuning for colour contrast: they are strongly UV-selective at lower contrasts, but become colour-opponent, i.e. additionally inhibited by green, for higher contrasts.”

We now add further explanation about the nature of the colour opponency:

“We confirmed the model’s predictions about G28’s nonlinear tuning for colour contrast using electrical recordings as described above. The example G28 (tSbC) cells shown in the figure exhibit similar nonlinear tuning in chromatic contrast space. The first example cell's firing rate and, consequently, the tuning curve peak for UV ON-green OFF stimuli (top left in panel e; upper row, second from left in f), and are lower for UV ON-green ON stimuli (top right in panel e; upper row, far right in f), reflecting the suppressive effect of green contrast on the cell's response.”

Figure 7i: The colors are hard to see and match to those in the rest of the figure. In addition, the gray bars and labels should be defined in the caption.

We changed that panel in Figure 7: instead of having all AUC curves in one plot, we now show the curves separately for the example RGC groups (vs. the curve of G28). We hope that this improves the readability of the panel.

Reviewer #2 (Recommendations for the authors):The paper has improved but still does not acknowledge its limitations sufficiently.

We thank the reviewer for appreciating our revisions.

1. With respect to the difference between the MEI and the linear receptive field, there was no misunderstanding in the previous review, and the current manuscript still does not acknowledge the limitations of the MEI analysis. This point could have been made more explicit in the previous review, and so the following is an attempt to do so.By analogy, in a topographic map of the world, the MEI is the location of Mt. Everest. The linear receptive field is a plane fit to the direction of the steepest ascent fit to the world map, additionally including an average slope. A Linear-Nonlinear model allows variation in height along that single direction. But a full understanding between the relationship between change in spatial position and height requires the full two-dimensional map.In the retina, and in a CNN model of the retina, parallel rectified pathways create sensitivity to many more different directions in stimulus space, commonly referred to as features. Each separate interneuron with a threshold potentially creates a different feature or dimension, with firing rate and sensitivity varying in each point of the high-dimensional space spanned by the set of features.Linear-Nonlinear models and MEIs only encode sensitivity to a single feature. However, multiple distinct features are required to produce a wide set of important phenomenon, including nonlinear subunits that cause sensitivity to fine textures, direction selectivity for both light and dark objects (On-Off), object motion sensitivity, latency coding, the omitted stimulus response, responses to motion reversal, and pattern adaptation (reviewed in Gollisch and Meister, 2010). All of these properties rely on multiple distinct features and their interactions. Any analysis based solely on an MEI necessarily abandons consideration of how sensitivity along these different features combines to produce a computation. The new Figure 6c, which examines two stimulus directions near the MEI is an improvement, but it is only two directions in one region of stimulus space.There is substantial concern that readers will misinterpret the 'Most Exciting' input to mean the 'Most Important'. It would therefore serve the field for a clear statement to be made that analysis of MEIs alone (1) will not capture interactions of multiple nonlinear neural pathways, (2) therefore will not capture nonlinear interactions between multiple stimulus features, and (3) will consequently not explain the set of phenomena that rely on these neural pathways and stimulus features.In response to the authors rebuttal, to clarify the nonlinear analysis that one might do beyond the analysis of MEIs to identify multiple dimensions, these would include Spike-Triggered Covariance (Fairhall et al., 2006), Maximally Informative Dimensions (Sharpee et al., 2003), Maximum Noise Entropy (Globerson et al., 2009), Nonnegative Matrix Factorization (Liu et al., 2017), proximal algorithms (Maheswaranathan et al., 2018), and model reduction approaches to reduce to the CNN model to fewer dimensions (Maheswaranathan et al., 2023) among others.

We thank the reviewer for clarifying the nature of their concerns. We did not mean to imply either that MEIs are the only approach, somehow the best or provide a complete picture of neuronal tuning. It is one method among many that can reveal nonlinear tuning properties and we used it to reveal a novel such tuning property. In the revised version of the manuscript, we have carefully rephrased all places where this impression might have been created. Specifically:

We rewrote parts of the Introduction and the Discussion to more clearly position the MEI approach as one of many potential analysis approaches, point out its limitations and discuss other approaches as well.

We added further guidance on how to interpret MEIs (as compared to Linear Receptive Fields) in the Results section.

2. It should be made explicit for each analysis whether it is based on measured responses or stimuli presented to the model. In Figure 7, a proper signal detection analysis cannot be performed without accounting for noise on single trials. It is unclear whether analyzed responses are directly from data, or from stimuli presented to the model. For extrapolation to different speeds, these are clearly from new stimuli presented to the model. It should be stated that without accounting for measured noise, ROC analyses will overestimate detection of ground-sky transitions.

Thank you for catching that this was not clear everywhere. We added labels to Figure 7 making explicit, which results are based on measured responses and which ones are based on model predictions. This is also clearly stated in the text.

We now additionally state that without accounting for measured noise, ROC analyses will overestimate detection of ground-sky transitions. Still, assuming roughly the same noise level across RGC groups, we can consider the ROC results based on model predictions as an upper bound on the expected detection performance for each RGC group. Therefore, we can interpret relative performance across RGC groups.